# SeedPrints: Fingerprints Can Even Tell Which Seed Your Large Language Model Was Trained From

**Yao Tong**[1][*]   **Haonan Wang**[1][*]   **Siquan Li**[2]   **Kenji Kawaguchi**[1]   **Tianyang Hu**[2][†]
[1]National University of Singapore   [2]The Chinese University of Hong Kong, Shenzhen
[1]{yaotong, haonan.wang}@u.nus.edu   kenji@comp.nus.edu.sg
[2]{lisiquan, hutianyang}@cuhk.edu.cn

## ABSTRACT

Fingerprinting Large Language Models (LLMs) is essential for provenance verification and model attribution. Existing fingerprinting methods are primarily evaluated after fine-tuning, where models have already acquired stable signatures from training data, optimization dynamics, or hyperparameters. However, most of a model's capacity and knowledge are acquired during pretraining rather than downstream fine-tuning, making large-scale pretraining a more fundamental regime for lineage verification. We show that existing fingerprinting methods become *unreliable* in this regime, as they rely on post-hoc signatures that only emerge after substantial training. This limitation contradicts the classical Galton notion of a fingerprint as an intrinsic and persistent identity. In contrast, we propose a stronger and more intrinsic notion of LLM fingerprinting: **SeedPrints**, a method that leverages random initialization biases as persistent, seed-dependent identifiers present even before training begins. We show that untrained models exhibit reproducible prediction biases induced by their initialization seed, and that these weak signals remain statistically detectable throughout training, enabling high-confidence lineage verification. Unlike prior techniques that fail during early pretraining or degrade under distribution shifts, **SeedPrints** remains effective across all training stages, from initialization to large-scale pretraining and downstream adaptation. Experiments on LLaMA-style and Qwen-style models demonstrate seed-level distinguishability and enable birth-to-lifecycle identity verification. Evaluations on large-scale pretraining trajectories and real-world fingerprinting benchmarks further confirm its robustness under prolonged training, domain shifts, and parameter modifications.

Together, our results show that initialization itself imprints a unique and persistent identity on LLMs, forming a true "Galtonian" fingerprint. Code is available at https://github.com/YnezT0311/SeedPrints.

## 1 INTRODUCTION

LLM fingerprints have recently been proposed as a tool to identify, attribute, and trace LLMs by examining their observable behaviors (Pasquini et al., 2024; Xu et al., 2024; Yoon et al., 2025; Zhang et al., 2024; Zeng et al., 2024). Such methods aim to provide model owners with a verifiable link between a suspicious model and its putative original, enabling detection of model theft or unauthorized reuse (Yoon et al., 2025; Zhang, 2025).

Much of this literature explicitly borrows the metaphor of biological fingerprints from Francis **Galton**'s *Finger Prints* (1892) (Galton, 1892):

> *"A fingerprint is the pattern formed by friction-ridge skin on the fingertips;*
> *this ridge configuration is individually unique and essentially permanent across an*
> *individual's lifetime."*

---

[*]Equal contribution.
[†]Correspondence to Tianyang Hu.

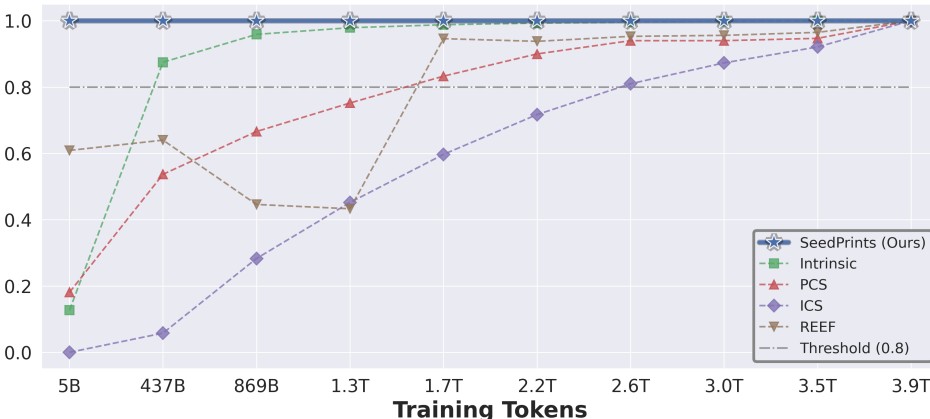

Figure 1: **Existing fingerprinting methods fail to detect model lineage during early pre-training.** We compare five methods on OLMo-2-7B checkpoints spanning 5B to 3.9T training tokens, each tested against the final checkpoint. The y-axis shows the similarity score (higher indicates a stronger lineage signal); the dashed line marks the 0.8 detection threshold. While all baselines degrade and fall below the threshold at early checkpoints, **SeedPrints** achieves perfect detection ($p \ll 0.001$, plotted as $1 - p$) from the very first checkpoint onward.

The analogy suggests that an effective fingerprint should be both unique and persistent, present from the very moment of a model's "birth" at initialization. Yet most existing so-called fingerprinting approaches fall short of this standard (Pasquini et al., 2024; Xu et al., 2024; Yoon et al., 2025; Zhang et al., 2024; Zeng et al., 2024; Zhang, 2025; Luan et al., 2025; Tsai et al., 2025; Alhazbi et al., 2025). As shown in Figure 1, existing methods fail to reliably detect lineage during early pretraining. This failure arises because these methods are defined only after models have fully converged, e.g., by extracting patterns from parameters or generated text. As a result, the separability they achieve reflects not a birthmark of the model itself, but an imprint of surface-level training components such as data signatures, optimization dynamics, or hyperparameters. For example, in Section 5.1, existing baselines often fail to correctly identify lineage when the training data distribution changes significantly. Such methods, therefore, function more as post hoc identifiers than as **Galtonian fingerprints**—those innate, immutable marks that accompany a model from its very beginning.

In this work, we propose a stricter notion of LLM fingerprinting as an intrinsic property present at *initialization* and detectable at *any* time of the subsequent training. Our key contributions are:

- **Identifying a fundamental limitation of existing methods.** We show that prior fingerprinting approaches fail to reliably detect lineage during early pretraining and can be easily misled under substantial shifts in the training data distribution. (Section 5.1)
- **Discovering and leveraging initialization-driven fingerprints.** We uncover that untrained models exhibit seed-dependent biases in their internal representations, forming a weak but persistent identity signal present from initialization. (Section 3) We then introduce **SeedPrints**, a method that isolates these initialization-born fingerprints and remains more robust to confounding factors such as data distribution, training duration, and optimization dynamics. (Section 4)
- **Extensive empirical validation.** We demonstrate that our method distinguishes models differing only in initialization seed, even under identical training pipelines and data orders, and remains persistent under continual training across diverse datasets. (Section 5.1)
- **Strong performance in realistic settings.** Our method reliably detects lineage throughout large-scale pretraining trajectories (e.g., OLMo-2-7B Stage 1), without early-stage failure. It also performs strongly in large-scale fine-tuning settings (e.g., up to 700B tokens) and practical deployment scenarios on *LeaFBench* benchmark (Shao et al., 2025), which includes 65 models spanning 7 mainstream model families and 6 transformations (e.g., instruction tuning, fine-tuning, PEFT, quantization, model merging, and distillation). In these settings, our method matches the strongest baseline (near-perfect) while substantially outperforming all others (Sections 5.2 and 5.3).

## 2   RELATED WORK

LLM protection broadly falls into two families: (i) *watermarking / active fingerprinting* methods that *insert* an identifiable signature into a model or its outputs (Xu et al., 2024; Nasery et al.; Tsai et al.,

2025; Sander et al., 2024), and (ii) *passive fingerprinting* methods that *extract* a signature from a model's pre-existing behaviors without modifying it (Yoon et al., 2025; Zhang et al., 2024; Alhazbi et al., 2025; Pasquini et al., 2024; Suzuki et al., 2025; Zhang, 2025).

**Watermarking and Active Fingerprinting.** Active approaches deliberately implant verifiable identifiers for later ownership checks. Classic techniques include backdoor attacks (Adi et al., 2018; Li et al., 2019; Zhang et al., 2018), digital signatures and hash functions (Guo & Potkonjak, 2018; Zhu et al., 2020). For language models, two common forms are: *text watermarks*, which bias generation or insert predefined patterns to encode hidden information (Kirchenbauer et al., 2023; Xu et al., 2024; Nasery et al.); and *model weight watermarks*, which embed identifiers into parameters or link them to secret triggers through fine-tuning (Luan et al., 2025; Li et al., 2023). Although backdoor-style fingerprints are relatively straightforward and may persist after moderate fine-tuning (e.g., Dasgupta et al. 2024), these invasive schemes require control of the training process, making them unsuitable for retroactively marking third-party models.

**Passive LLM Fingerprinting.** In contrast, passive fingerprinting identifies models by analyzing their intrinsic properties without any modification. Passive fingerprinting techniques vary by model access. With **white-box access**, signatures are extracted from model weights, leveraging intrinsic properties like the distribution of attention matrices (Yoon et al., 2025), the kernel alignment of internal representations (Zhang et al., 2024), or the stable direction of parameter vectors. In the **black-box setting**, fingerprinting relies on analyzing input-output behavior. These methods use crafted queries (Pasquini et al., 2024), unique prompt-response pairs (Tsai et al., 2025), stylometry (Alhazbi et al., 2025) or iterative prompting–response games (Iourovitski et al., 2024) to identify a model, though they can be less robust to fine-tuning. However, these methods define their signatures *post hoc*. Specifically, they identify emergent properties from a completed training process, rather than the innate, "*Galtonian*" fingerprints present from random initialization that our work seeks to discover. As a result, *they are not designed for lineage verification during large-scale pretraining*, where such post-training signals have not yet emerged.

## 3 BIASES ORIGINATING FROM INITIALIZATION PERSIST AFTER TRAINING

In this section, we present our key observation that language models exhibit strong prediction biases originating from the random initialization seed, and such biases remain detectable even after training.

**Initialization induces seed-specific bias profiles.** We evaluate a LLaMA-2–style model initialized with seed 123 on 10,000 random input sequences of length 1,024, where tokens are sampled uniformly from the vocabulary. In Figure 2 (*Left*), we plot the frequency with which each output token (upper, from the final logits) and each hidden dimension (lower, from the last-layer representations) attains the minimum value across the 10,000 random trials. The pronounced non-uniform shape in both plots indicates the presence of extreme output preference bias patterns. Repeating the experiment across different initialization seeds shows that while the overall magnitude of bias remains stable, the specific argmin dimensions depend on the seed (Figure 2, *Lower Right*). These observations are consistent with recent findings that biases in randomly initialized models arise from inter-sequence representation contraction, driven by asymmetric nonlinear activations in MLP blocks and further amplified by self-attention (Li et al., 2026). Under this view, different initialization seeds induce different contraction directions, resulting in distinct argmin patterns across output dimensions.

**Training preserves initialization-born bias** Although training substantially changes output magnitudes, the relative preference over output dimensions induced at initialization is not entirely lost. We train the previously initialized model (seed 123) for one epoch on OpenWebText (Gokaslan et al., 2019) and evaluate intermediate checkpoints. We focus on the $m = 50$ output dimensions that are most frequently assigned minimum values by the initialized model. For each checkpoint, we compute the correlation between its responses and those of the initialized model across random inputs, restricted to these dimensions. Intuitively, this measures how similarly the checkpoint and the initialized model rank or disfavor these dimensions across inputs. As a baseline, we compute the same correlation with other independent initialized models (seed 1000). As shown in Figure 2 (Upper Right), although the absolute correlation values are small, correlations for models from the same initialization are consistently shifted upward relative to the independent baseline, and stabilize rather than decaying to zero. Overall, this indicates that initialization leaves a weak but statistically detectable bias signal that persists throughout training.

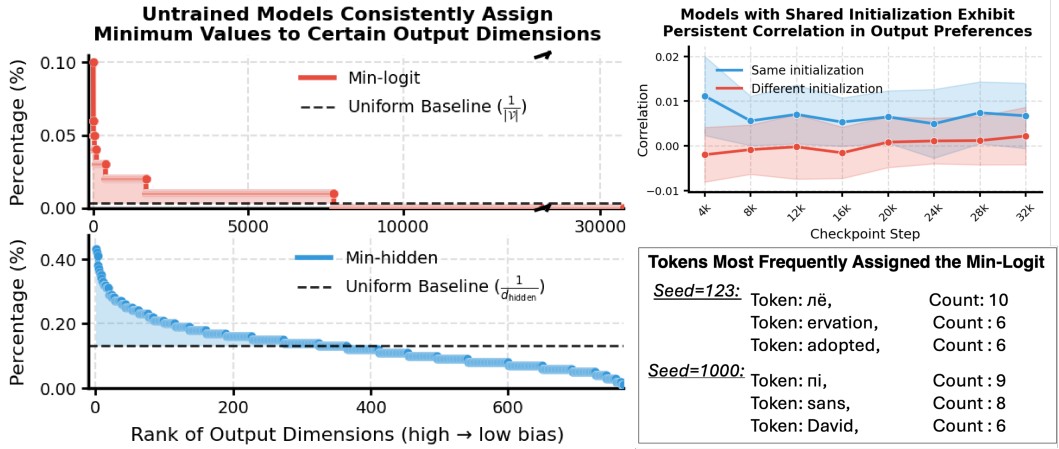

Figure 2: **Initialization-born output bias persists through training.** *Left:* Given completely random inputs, the outputs of a randomly initialized LLaMA-2–style model are far from uniform, but instead exhibit clear bias: certain dimensions are disfavored by the model (i.e., they frequently receive the minimum value across random inputs). Such extreme bias appears both in the logits (top, red) and in the final hidden representations (bottom, blue). The dashed line shows the expected frequency under a uniform distribution. The arrows in the top panel indicate a broken x-axis that omits low-frequency tail ranks. *Upper Right:* During training, models remain weakly correlated in their output bias across inputs with the base model they are trained from. This correlation distribution is consistently shifted upward compared to correlations computed with independently initialized base models, indicating that initialization-born bias leaves a measurable signal that persists through training. *Lower Right:* The specific dimensions that exhibit strong bias depend on the initialization seed; here we illustrate this using tokens that most frequently receive the minimum logit.

## 4 ALGORITHM

Section 3 shows that models from the same initialization lineage retain weak but consistent agreement on a subset of output dimensions (e.g., those most disfavored by the untrained model under random inputs). Our goal is therefore to uncover a set of dimensions that encode such bias signals from any two given models (without access to the true initialized model), and test whether their agreement is statistically significant. The overall procedure is summarized in Algorithm 1, and we describe each component below.

**Extract identity dimensions between any two models.** Let $X = \{x_i\}_{i=1}^n$, where each $x_i \in \mathbb{R}^{\ell \times d}$ denotes a random sequence of length $\ell$. Each random input $x_i$ can be instantiated either by uniformly sampling $\ell$ token IDs from the vocabulary, or by directly sampling $\ell$ independent vectors from a $d$-dimensional isotropic Gaussian distribution. For any model $g$, we define its mean response vector as $\bar{g} := \frac{1}{n} \sum_{i=1}^n g(x_i) \in \mathbb{R}^{d_{\text{out}}}$, where $d_{\text{out}}$ denotes the output dimensionality. Here, $g(x_i)$ can denote either the model's logits or its final-layer representations. We then identify its disfavored dimensions as the set of bottom-$m$ coordinates of $\bar{g}$:

$$\mathcal{M}_g := \underset{J \subseteq \{1, \ldots, d_{\text{out}}\}, \, |J| = m}{\arg\min} \sum_{j \in J} \bar{g}_j.$$

We use the mean response vector to extract bias signals, as it provides a more stable and noise-robust alternative to per-sample argmin frequency (i.e., counting how often each dimension attains the minimum value, as illustrated in Figure 2), while also performing well empirically. We provide a formal comparison in Appendix C.1.

Given two models $f$ and $f'$, we define their *identity dimensions* as the intersection $\mathcal{S} := \mathcal{M}_f \cap \mathcal{M}_{f'}$. Intuitively, if two models share the same initialization lineage, they should independently identify a similar set of disfavored dimensions, leading to non-trivial overlap in their bottom-$m$ sets. This intersection-based criterion therefore acts as a form of mutual verification: it suppresses spurious correlations arising from architectural similarity or shared training dynamics, and isolates bias signals that are more likely to originate from shared initialization lineage. When one of the models is closer to initialization, its disfavored dimensions can also be used directly as an anchor.

---

**Algorithm 1** Distribution Correlation Test on Identity Dimensions

---

**Require:** base model $f$, suspicious model $f'$; random inputs $X = \{x_i\}_{i=1}^n$; fingerprint size $m$;
 significance level $\alpha$

    ▶ *Step 1: Localize biased dimensions*
1: Compute average outputs $\bar{f}$, $\bar{f}'$ over $X$
2: $\mathcal{M}(f)$, $\mathcal{M}(f') \leftarrow$ bottom-$m$ dimensions of $\bar{f}$, $\bar{f}'$
3: $\mathcal{S} \leftarrow \mathcal{M}(f) \cap \mathcal{M}(f')$                                          (identity dimensions)
    ▶ *Step 2: Form correlation distribution*
4: Normalize $f(x_i)_{\mathcal{S}}$, $f'(x_i)_{\mathcal{S}}$ for each $x_i \in X$ across identity dimension
5: **for** each $s_j \in \mathcal{S}$ **do**
6:     $\tau_j \leftarrow \mathrm{KendallTau}\big(\{f(x_i)_{s_j}\}_{i=1}^n, \{f'(x_i)_{s_j}\}_{i=1}^n\big)$          (correlation in in input-wise preferences)
7: **end for**
8: $\mathcal{T} \leftarrow \{\tau_j\}_{j=1}^{|T|}$
    ▶ *Step 3: Hypothesis test against null*
9: Construct $\mathcal{T}_{\mathrm{null}}$ by applying the same pipeline to two Gaussian matrices $Y^{(1)}, Y^{(2)} \sim \mathcal{N}(0, I)^{N \times d_{\mathrm{out}}}$
10: Test $H_0 : \mathcal{T} = \mathcal{T}_{\mathrm{null}}$ vs. $H_1 : \mathcal{T} > \mathcal{T}_{\mathrm{null}}$
11: **Return** SameLineage $\leftarrow \mathbf{1}(p\text{-value} < \alpha)$

---

**Correlation statistics.** Restricting both models to the identity dimensions $S$, we compare their responses across random inputs.[1] For each $j \in S$, we compute a Kendall–Tau rank correlation:

$$\tau_j = \mathrm{KendallTau}(\{f(x_i)_j\}_{i=1}^n, \{f'(x_i)_j\}_{i=1}^n).$$

This yields a set of correlation statistics $T = \{\tau_j\}_{j \in S}$, which captures the distribution of agreement between the two models across identity dimensions. As observed in Section 3, models from the same lineage exhibit distributionally higher agreement on these dimensions compared to unrelated models.

**Hypothesis testing via null distribution.** Since the correlation distributions for same-lineage and different-lineage models can overlap, a simple threshold on individual statistics is insufficient. Instead, we test whether the observed correlations are significantly larger than what would be expected under the null hypothesis of no lineage relation. Ideally, this null should be estimated from the correlation distribution between independent models as in Figure 2. However, obtaining such a baseline empirically requires comparisons across many independently initialized models, which is is computationally expensive. As shown in Appendix B, the null correlation distribution between independently initialized models is well approximated by a Gaussian distribution. This naturally motivates a cheaper simulation-based alternative: constructing an empirical null distribution using Gaussian surrogates passed through the same pipeline, as summarized in Algorithm 1.

While this simulation-based null already provides a strong empirical approximation, it still introduces unnecessary randomness and computational overhead: its precision is limited by the number of simulation trials, results can vary across runs, and comparing two empirical distributions typically requires additional assumptions from the adopted test, such as the one-sided t-test or the Mann–Whitney U test. Since the null distribution of Kendall–Tau (under weak independence) is available in closed form, we further replace this simulation-based procedure with an analytical test, while also providing both empirical and analytical implementations in our code repository. Specifically, under the null hypothesis, each Kendall–Tau correlation is approximately zero-mean with known variance $\sigma^2$. By the central limit theorem, as the number of identity dimensions $|S|$ increases, the distribution of $\bar{\tau} = \frac{1}{|S|} \sum_{j \in S} \tau_j$ is well approximated by $\bar{\tau} \sim \mathcal{N}\left(0, \frac{\sigma^2}{|S|}\right)$. While softmax normalization introduces weak dependencies across dimensions, these effects are mitigated by the use of a relatively high temperature ($T = 10$), making the independence assumption a reasonable approximation in practice. We empirically validate that this Gaussian approximation closely matches the distribution obtained from the full pipeline (Appendix B). We therefore compute the one-sided z-score $z = \frac{\bar{\tau}}{\sigma/\sqrt{|S|}}$, and the corresponding p-value $p = 1 - \Phi(z)$, where $\Phi$ is the standard normal cumulative distribution function. We reject the null hypothesis and declare shared lineage if $p < 0.01$.

**Practical instantiation.** Model outputs can be taken as either logits or hidden representations. While both choices are consistent with our formulation, using logits requires significantly more

---

[1]Before computing the correlations, we apply a row-wise softmax normalization to each sample's output vector. This removes scale differences across different models and dimensions while preserving the relative ordering that carries identity information for rank-based statistics such as Kendall–Tau. A relatively high temperature is used to avoid degenerate one-hot behavior.

Table 1: Comparison of fingerprint behaviors between models initialized with different seeds.

| Model Pairs | $p$-value ($> 0.01$) |
|---|---|
| $s_{42}$ vs. $s_{2000}$ | 0.116 |
| $s_{123}$ vs. $s_{42}$ | 1.000 |
| $s_{1000}$ vs. $s_{123}$ | 0.327 |
| $s_{2000}$ vs. $s_{1000}$ | 1.000 |

Table 2: Trained models share the same fingerprint behaviors as their initialization ($p$-value $< 0.01$).

| Model Pairs | SeedPrints $p$-value | Baselines Intrinsic | REEF | PCS | ICS |
|---|---|---|---|---|---|
| $s_{42}^{init}$ vs. $s_{42}^{base}$ | 3.37e-26$^{\checkmark}$ | -0.021$^{\times}$ | 0.375$^{\times}$ | 0.580$^{\times}$ | 0.196$^{\times}$ |
| $s_{123}^{init}$ vs. $s_{123}^{base}$ | 3.16e-33$^{\checkmark}$ | 0.149$^{\times}$ | 0.369$^{\times}$ | 0.581$^{\times}$ | 0.188$^{\times}$ |
| $s_{1000}^{init}$ vs. $s_{1000}^{base}$ | 1.12e-21$^{\checkmark}$ | -0.252$^{\times}$ | 0.381$^{\times}$ | 0.581$^{\times}$ | 0.188$^{\times}$ |
| $s_{2000}^{init}$ vs. $s_{2000}^{base}$ | 3.70e-31$^{\checkmark}$ | -0.337$^{\times}$ | 0.331$^{\times}$ | 0.581$^{\times}$ | 0.188$^{\times}$ |

random inputs to reliably estimate bias patterns as the vocabulary size increases, leading to substantial computational overhead (see Appendix C.2 for theoretical analysis). To improve efficiency and stability, we therefore use final-layer hidden states as the default model output. The resulting p-value provides a direct statistical test for lineage, without relying on heuristic similarity thresholds.

## 5 EXPERIMENTS

This section consists of two parts. In Section 5.1, we demonstrate that our method functions as a genuine fingerprint: (i) it enables *birth verification at the seed level*, and (ii) it remains *verifiable across the entire training lifecycle*. In contrast, existing baselines fail to support verification at early pretraining stages and primarily rely on data-dependent signals, which break down under significant distribution shifts. We report results using the analytical null with *hidden state* outputs, and include additional results based on the empirical null with both *logit* and *hidden state* outputs, evaluated using the one-sided $t$-test and Mann–Whitney $U$ test in the Appendix D.2. In Section 5.3, we evaluate all methods under practical infringement scenarios using *LeaFBench* (Shao et al., 2025). Across 65 distinct model instances spanning 6 representative post-development techniques, our method matches the best baselines in post-training settings and remains robust to diverse deployment transformations.

**Baselines** We mainly consider four passive fingerprinting baselines (weight- or representation-based). *Intrinsic fingerprint* (Yoon et al., 2025) (or *PDF* in some papers) compares models via the similarity of the layerwise standard-deviation profiles of attention parameters. *REEF* (Zhang et al., 2024) computes centered-kernel-alignment (CKA) similarity between feature representations from the same samples across two models. *PCS* and *ICS* (Zeng et al., 2024) (or collectively as *HuRef* in some papers) are weight-similarity methods: PCS flattens all parameters and measures cosine similarity; ICS forms invariant terms from the weights and measures cosine similarity on those invariants. In Section 5.3, we additionally include a (weaker) gradient-based fingerprint *Gradient* (Shao et al., 2025). Following Zhang et al. (2024), we use a **0.8 similarity threshold** for binary decisions.

Note, in all experiment tables, cell colors indicate lineage: with green denotes models from the same source, and red denotes different sources. For example, $s_{42}^{init}$ vs. $s_{42}^{base}$ compares a model initialized with seed 42 and its continued-pretrained counterpart, hence green. By contrast, $s_{2000}^{init}$ vs. $s_{42}^{base}$ compares a seed-2000 initialization with a model trained from a seed-42 initialization, hence red. Additionally, $\checkmark$ denotes a correct detection, while $\times$ denotes an error.

### 5.1 BIRTH-TO-LIFECYCLE "BIOMETRIC" FINGERPRINTING

We train 12-layer, 12-head LLaMA-style models (Touvron et al., 2023) with RoPE (Su et al., 2021) and Qwen-style models (Team, 2024) from scratch. In the main paper, we present results for LLaMA-style models and defer those for Qwen-style models to Appendix D. The overall conclusions are consistent across the two.

**Different initialization seeds produce distinct fingerprints** Table 1 reports $p$-values from our correlation tests between pairs of models initialized with different random seeds (42, 123, 1000, and 2000). All $p$-values are consistently $> 0.01$, indicating that our method reliably distinguishes models trained from different seeds. This shows that distinct seeds yield distinct fingerprint behaviors, allowing models to be separated "at birth."

**Training preserves the initialization fingerprint.** Table 2 compares each initialization model $s^{init}$ with its descendant $s^{base}$ trained on the OpenWebText dataset (Gokaslan et al., 2019) ($\approx$10B tokens). Across all seed–model pairs, $p$-values are consistently $< 0.01$, indicating that their bias profiles

Table 3: The same dataset and training order do not shape fingerprint behaviors to be identical across different initializations.

| Model Pair | $p$-value ($> 0.01$) |
|---|---|
| $s_{42}^{init}$ vs. $s_{123}^{base}$ | 0.573 |
| $s_{123}^{init}$ vs. $s_{2000}^{base}$ | 0.724 |
| $s_{2000}^{init}$ vs. $s_{1000}^{base}$ | 0.109 |
| $s_{1000}^{init}$ vs. $s_{42}^{base}$ | 0.883 |

Table 4: Fingerprint persistence under continual training on diverse datasets (base model: seed 1000, corpus `openwebtext`).

| Setting | SeedPrints | Baselines | | | |
|---|---|---|---|---|---|
| Continual corpus (seed) | $p$-value | Intrinsic | REEF | PCS | ICS |
| `TinyStories` (1000) | $\approx 0$ ✓ | 1.000 ✓ | 0.759 ✗ | 0.999 ✓ | 0.996 ✓ |
| `TinyStories` (123) | 1.000 ✓ | 0.950 ✗ | 0.658 ✓ | 0.332 ✓ | 0.012 ✓ |
| `the_stack` (1000) | $\approx 0$ ✓ | 0.489 ✗ | 0.557 ✗ | 0.585 ✗ | 0.123 ✗ |
| `the_stack` (123) | 1.000 ✓ | 0.445 ✓ | 0.580 ✓ | 0.301 ✓ | 0.026 ✓ |

remain strongly correlated and thus share a common lineage. In short, the trained model inherits the same fingerprint as its initialization. We also evaluate baseline methods; without exception, they fail to distinguish across seeds, which in turn suggests their separability stems from training-induced artifacts rather than initialization.

**Identical data and order do not make fingerprints converge** In Table 3, all four "suspicious" models $s_i^{base}$ for $i \in \{42, 123, 100, 2000\}$ are trained on *exactly the same corpus (OpenWebText) and in the same data order* (we fix the training seed to lock the data order); the only difference lies in their initialization seeds $i$. Across all cross-seed pairs, $p$-values remain consistently $> 0.01$, in sharp contrast to the near-zero values in Table 2. That is, fingerprints remain seed-specific even under identical data and curriculum.

**Continual training on diverse datasets does not confound the fingerprint** The purpose of our earlier experiments is solely to demonstrate the strengths of our SeedPrints: it can act as a biometric fingerprint. From a copyright perspective, the inability of prior works to distinguish models with different seeds is not a weakness, since initialization seeds have no clear copyright status. The real fragility is that their attribution can be easily misled by *data distribution*, ***failing to recognize lineage when the training distribution shifts substantially during continued training***.

In Table 4, we continue training a base model (seed 1000, pretrained on OpenWebText (Gokaslan et al., 2019)) on two very different datasets: TinyStories (Eldan & Li, 2023) (synthetic children's stories) and The Stack (Kocetkov et al., 2022) (permissively licensed GitHub code). We compare (i) true descendants trained from the base, versus (ii) *distractors* derived from a different base model (seed 123). Since baseline methods are not sensitive to initialization differences, the distractor base is pretrained with a different data order on OpenWebText, before being continually trained on the *same* target corpus to form the distractor. The question is whether attribution methods can identify which descendant truly shares lineage with the base.

We find that prior baselines all fail under the code setting (The Stack), misclassifying true descendants as distractors. This indicates that they largely track domain similarity rather than lineage identity: TinyStories is closer in distribution to the pretraining corpus (OpenWebText), while The Stack diverges sharply; such a large distribution shift can easily bypass detection. In contrast, our method correctly attributes lineage across both corpora. Hence, our fingerprint is not a proxy for data distribution: it survives substantial domain shift and persists beyond the initial pretraining stage.

## 5.2 ALL-STAGE VERIFIABLE FINGERPRINTS

**Controlled pretraining trajectory.** Can our method reliably identify its offspring along the training trajectory? We first verify this via a controlled pretraining experiment on OpenWebText, where the initialization seed is known, by testing lineage at intermediate checkpoints, and observe consistent detection across all stages (Appendix D.2.1, Figure 5).

**Large-scale pretraining.** To evaluate lineage under realistic foundation model training, we further analyze ten checkpoints from the OLMo-2-7B (OLMo et al., 2024) Stage-1 run (5B → 3.9T tokens), where we treat the final checkpoint as the target model and all previous checkpoints as tested ancestors. Unlike the controlled OpenWebText setting, this stage spans heterogeneous corpora, longer optimization, and large-scale representation drift. We plot $1 - p$ for consistent directionality with similarity-based baselines. All $p$-values are $\ll 0.01$ and numerically close to zero.

In Figure 1, we find that **all existing baselines fail to verify lineage during early pretraining** (e.g., ICS$\approx 0$, PCS$< 0.3$ and REEF$< 0.6$ within the first **trillion** tokens), whereas SeedPrints remains

Table 5: Fingerprinting results under large-scale finetuning. Each row compares a target model against LLaMA-2-7B. **SeedPrints** reports the $p$-value from our correlation test ($< 0.01$ indicates a strong signal). Four baselines all report similarity scores (threshold $= 0.8$, higher = better).

| Model | # Tokens | SeedPrints $p$ | Intrinsic ↑ | REEF (↑) | PCS (↑) | ICS (↑) |
|---|---|---|---|---|---|---|
| Llama-2-finance-7B (Heenan, 2023) | 5M | $10^{-41955}$✓ | 1.0000✓ | 0.9950✓ | 0.9979✓ | 0.9952✓ |
| Vicuna-1.5-7B (Chiang et al., 2023) | 370M | $10^{-103043}$✓ | 1.0000✓ | 0.9985✓ | 0.9985✓ | 0.9949✓ |
| Wizardmath-7B (Luo et al., 2023) | 1.8B | $10^{-180550}$✓ | 1.0000✓ | 0.9979✓ | 1.0000✓ | 0.9994✓ |
| Meditron-7B (Chen et al., 2023) | 48B | $10^{-42566}$✓ | 0.9990✓ | 0.9978✓ | 1.0000✓ | 0.9817✓ |
| CodeLlama-7B (Meta AI, 2024) | 500B | $10^{-3552}$✓ | 0.9480✓ | 0.9947✓ | 0.6863× | 0.3369× |
| Llemma-7B (Azerbayev et al., 2023) | 700B | $10^{-5136}$✓ | 0.9470× | 0.9984✓ | 0.6682× | 0.2905× |

consistently verifiable and strengthens throughout training. These results highlight a critical gap in existing evaluations: **most prior work focuses on verifying lineage only after fine-tuning, where lineage detection is substantially easier.** However, our findings show that large-scale pretraining itself can dramatically reshape representations and strengthen lineage signals, potentially giving a *false sense of safety*. We argue that lineage verification must prioritize the early stages of pretraining when misuse is most difficult to detect, rather than relying solely on late-stage checkpoints.

**Large-scale finetuning.** We further compare our method with existing baselines under standard evaluations on finetuning stage. In particular, we test suspect models fine-tuned from Llama-2-7b (base model) with data volumes ranging from 5 million to 700 billion tokens. The suspects include diverse downstream variants such as Llama-2-finance-7b (Heenan, 2023), Vicuna-1.5-7b (Chiang et al., 2023), WizardMath-7b (Luo et al., 2023), Chinese-LLaMA-2-7b (Chen et al., 2023), Code-Llama-7b (Meta AI, 2024), and Llemma-7b (Azerbayev et al., 2023). Their fine-tuning data volumes are 5M, 370M, 1.8B, 13B, 500B and 700B tokens, respectively. As shown in Table 5, our method consistently maintains $p < 0.01$ across all settings.

## 5.3 ROBUSTNESS UNDER REALISTIC DEPLOYMENTS

Real-world deployments routinely apply *parameter-altering* adaptations to foundation models (e.g., fine-tuning, PEFT, quantization), which can weaken or distort fingerprint signals. Our method is designed to remain effective in this regime. To evaluate its reliability under such realistic conditions, we adopt *LeaFBench* (Shao et al., 2025), a benchmark for language-model copyright auditing.

**Across Source Models.** We consider two types of source models for auditing: *Pre-Trained (PT)* and *Instruction-Tuned (IT)*. For each source type, the task is to distinguish *derivative* models (obtained by post-training adaptations) from *independent* models (trained without using the source weights). This mirrors common auditing scenarios where one must test lineage claims for either a PT base or its IT variant. Our evaluation covers **65** models in total; the full list of model names and HuggingFace repositories is provided in Appendix E.

**Metrics.** We evaluate model detection performance using (i) the Area Under the ROC Curve (AUC) and (ii) the Kolmogorov–Smirnov (KS) statistic (Berger & Zhou, 2014), which measures the maximum separation between score distributions of derivative and independent models.

Our method produces per-model p-values from statistical tests, which differ fundamentally from the similarity scores used by baseline methods. While similarity scores are linear measures where differences have proportional meaning, p-values are tail probabilities indicating the rarity of an observation under the null hypothesis—they cannot be interpreted on a linear scale. To enable comparison with baseline methods that compute AUC from similarity scores, we convert our $p$-values into scores via $s = 1 - p$. This conversion is not fully aligned with the statistical meaning of $p$-values and may be suboptimal for our method, as small $p$-values change on an exponential scale while threshold sweeping is linear, making it hard to distinguish fine-grained differences. Nevertheless, our approach still achieves comparable or superior performance. For evaluation, we (i) compute AUC for all methods by sweeping thresholds over the scores $s$, and (ii) report the KS statistic as a threshold-free measure of distributional separability. This protocol ensures compatibility with baseline pipelines while preserving the statistical interpretability of our method. We do not report Manhattan distance, as absolute distances between $p$-values are not meaningful.

Note that, in most practical scenarios, calibrating an optimal similarity threshold is infeasible. While we report AUC for comparability, our method fundamentally differs from prior methods: it provides

Table 6: Performance comparison of LLM fingerprinting methods across different source models. "PT" and "IT" refer to using the pre-trained models and instruction-tuned models as source models, respectively. Per-family breakdown is shown for the 6 families that have both PT and IT models. Note that the "Overall" scores are computed over all test model pairs, not averaged across families.

| Method | Metric | Qwen2.5-7B | | Qwen2.5-14B | | Llama3.1-8B | | Mistral-7B | | Gemma2-2B | | Llama2-7B | | Overall |
|---|---|---|---|---|---|---|---|---|---|---|---|---|---|---|
| | | PT | IT | PT | IT | PT | IT | PT | IT | PT | IT | PT | IT | |
| REEF | AUC ↑ | 0.796 | 0.862 | 0.947 | 0.913 | 1.000 | 0.999 | 0.997 | 0.997 | 0.963 | 0.968 | 0.750 | 0.706 | **0.915** |
| | KS Statistic ↑ | 0.492 | 0.633 | 0.875 | 0.735 | 0.987 | 0.987 | 0.975 | 0.980 | 0.950 | 0.837 | 0.750 | 0.694 | **0.739** |
| Gradient | AUC ↑ | 0.657 | 0.743 | 0.763 | 0.789 | 0.897 | 0.904 | 0.872 | 0.682 | 1.000 | 1.000 | 0.650 | 0.768 | **0.801** |
| | KS Statistic ↑ | 0.354 | 0.423 | 0.500 | 0.563 | 0.646 | 0.705 | 0.725 | 0.500 | 1.000 | 1.000 | 0.275 | 0.479 | **0.508** |
| ICS | AUC ↑ | 1.000 | 1.000 | 0.997 | 0.997 | 1.000 | 1.000 | 1.000 | 1.000 | 1.000 | 1.000 | 1.000 | 1.000 | **0.994** |
| | KS Statistic ↑ | 1.000 | 1.000 | 0.975 | 0.979 | 1.000 | 1.000 | 1.000 | 1.000 | 1.000 | 1.000 | 1.000 | 1.000 | **0.943** |
| Intrinsic | AUC ↑ | 1.000 | 1.000 | 1.000 | 1.000 | 1.000 | 1.000 | 1.000 | 1.000 | 1.000 | 1.000 | 1.000 | 1.000 | **0.994** |
| | KS Statistic ↑ | 1.000 | 1.000 | 1.000 | 1.000 | 1.000 | 1.000 | 1.000 | 1.000 | 1.000 | 1.000 | 1.000 | 1.000 | **0.989** |
| SeedPrints | AUC ↑ | 0.992 | 0.994 | 0.988 | 0.990 | 1.000 | 1.000 | 1.000 | 1.000 | 1.000 | 1.000 | 1.000 | 1.000 | **0.992** |
| | KS Statistic ↑ | 0.985 | 0.987 | 0.975 | 0.980 | 1.000 | 1.000 | 1.000 | 1.000 | 1.000 | 1.000 | 1.000 | 1.000 | **0.986** |

a direct statistical test. **Given any pair of models, our approach outputs a $p$-value that enables a definitive decision on whether they share the same lineage, without relying on tunable thresholds**, ensuring reliable verification in practice.

**Parameter-Altering Techniques.** To assess fingerprinting robustness, we evaluate models produced via (1) Instruction tuning (Instruct), (2) General-purpose fine-tuning (Finetune), (3) Parameter-efficient fine-tuning (PEFT), (4) Quantization, (5) Model merging (Merge), and (6) Distillation.

### 5.3.1 FAMILY-WISE EFFECTIVENESS

There are seven model families in total. For each family, we treat both its pretrained base model (PT) and its instruction-tuned model (IT) as separate source models and compare them with all other models in the pool of 65. The results are reported in Table 6. We omit the TinyLlama-1.1B families in Table 6, as they constitute trivial cases for all methods. Note that each column corresponds to one family (AUC is then computed over the corresponding tests, using a family-calibrated threshold), whereas the overall score is computed across all test pairs with a global threshold. Thus, the overall score is not equal to the average of the per-family scores. All three methods—*Intrinsic*, *ICS*, and our *SeedPrints*—are essentially saturated across all alterations (overall AUC of 0.994, 0.994, and 0.992, respectively; KS of 0.989, 0.943, and 0.986). For these methods, the family-wise scores are, in most cases, the full score. By contrast, the remaining two baselines, *REEF* and *Gradient*, perform substantially worse. These results indicate that, while our method is primarily designed to capture intrinsic seed-level fingerprints, it effectively functions as a biometric-like fingerprint that provides reliable and persistent identity tracking across diverse model families.

### 5.3.2 ALTERATION-WISE ROBUSTNESS

Across alteration types, *Instruct* is consistently the easiest case (even the weakest baseline, Gradient, achieves AUC 0.895), whereas *Finetune* and *Merge* are overall more challenging for all methods, e.g., for *Finetune*, no method achieves perfect performance. The effects of *PEFT*, *Quantization*, and *Distillation* are more mixed: REEF shows notable drops under *Quantization* ($0.996 \rightarrow 0.871$) and *Distillation* ($0.996 \rightarrow 0.858$), while remaining resilient to *PEFT* (0.994); conversely, *PEFT* and *Quantization* are more challenging for Gradient ($0.895 \rightarrow 0.776/0.756$). The other three stronger methods are mostly robust to all transformations (all scores $> 0.9$). SeedPrints drops slightly to 0.959 on *Merge*, failing on a single merge model out of seventeen. While this is only one misclassification, it suggests that weight-space interpolation with unrelated model weights can potentially dilute the initialization signal below the detection threshold. Averaged across all six alteration types, Intrinsic still achieves the highest overall AUC, followed closely by ICS and SeedPrints.

### 5.4 ABLATION STUDY ON HYPERPARAMETERS

Our method has three hyperparameters: the number of random input sequences $n$, the length of each random sequence $l$, and the fingerprint size $m$. The first two control the overall size of random inputs, while $m$ specifies how many low-mean coordinates are used for testing. Throughout the main experiments in Sections 5.2 and 5.3, we set $n=2000$, $l=1024$, and $m=0.1d_{\text{out}}$, where $d_{\text{out}}$ denotes the

Table 7: Performance of LLM fingerprinting methods across different parameter-altered techniques.

| Method | Metric | Instruct | Finetune | PEFT | Quantization | Merge | Distillation | Overall Avg |
|--------|--------|----------|----------|------|--------------|-------|--------------|-------------|
| REEF | AUC ↑ | 0.996 | 0.872 | 0.994 | 0.871 | 0.954 | 0.858 | **0.924** |
| | KS Statistic ↑ | 0.949 | 0.637 | 0.940 | 0.812 | 0.851 | 0.615 | **0.801** |
| Gradient | AUC ↑ | 0.895 | 0.729 | 0.776 | 0.756 | 0.832 | 0.887 | **0.812** |
| | KS Statistic ↑ | 0.756 | 0.441 | 0.469 | 0.586 | 0.519 | 0.679 | **0.575** |
| ICS | AUC ↑ | 1.000 | 0.987 | 1.000 | 1.000 | 0.981 | 1.000 | **0.995** |
| | KS Statistic ↑ | 1.000 | 0.925 | 1.000 | 1.000 | 0.941 | 1.000 | **0.978** |
| Intrinsic | AUC ↑ | 1.000 | 0.980 | 1.000 | 1.000 | 1.000 | 1.000 | **0.997** |
| | KS Statistic ↑ | 1.000 | 0.971 | 1.000 | 1.000 | 1.000 | 1.000 | **0.995** |
| SeedPrints | AUC ↑ | 1.000 | 0.995 | 0.990 | 1.000 | 0.959 | 1.000 | **0.991** |
| | KS Statistic ↑ | 1.000 | 0.990 | 0.980 | 1.000 | 0.941 | 1.000 | **0.985** |

hidden state dimensionality, making the method adjustable for different open-source models. Here, we vary each hyperparameter while fixing the remaining ones to their default values (i.e., $n = 2000$, $m = 400$, $l = 768$), and report empirical accuracy and false positive rates under a fixed significance level $\alpha = 0.01$ in Table 8.

We observe that increasing $n$ and $l$ consistently improves identification performance. This trend is explained by our theoretical analysis in Section C.2, which shows that to accurately recover the underlying population-level bias, the required number of queries must satisfy $n = \Omega\left(\frac{\log d_{\text{out}}}{\gamma^2}\right)$, where $\gamma$ is the bias margin separating biased coordinates from the remaining dimensions. This means a target output dimension requires a minimal number of samples for accurate detection, explaining why smaller $n$ leads to suboptimal performance. On the other hand, $\gamma$ increases with $l$ due to the accumulation effects, meaning longer sequences strengthen separation. Therefore, under a fixed (insufficient) number of sequences, increasing $l$ reduces the required query count and improves detection.

Table 8: Ablation study under significance level $\alpha = 0.01$.

| Ablation on $n$ | 200 | 500 | 2000 |
|-----------------|-----|-----|------|
| ACC | 0.7368 | 0.7778 | 0.9375 |
| Empirical FPR | 0 | 0 | 0 |

| Ablation on $l$ | 512 | 768 | 1024 |
|-----------------|-----|-----|------|
| ACC | 0.9048 | 0.9375 | 1.0000 |
| Empirical FPR | 0.04 | 0 | 0 |

| Ablation on $m$ | 200 | 400 | 800 |
|-----------------|-----|-----|------|
| ACC | 0.9375 | 0.9375 | 0.875 |
| Empirical FPR | 0.04 | 0 | 0.1463 |

The effect of $m$ is non-monotonic. When $m$ is too small, the selected coordinates are unstable due to insufficient empirical observations to form a reliable distribution. When $m$ is too large, the inclusion of many irrelevant coordinates introduces noise that shrinks $\gamma$, which would require quadratically more samples to compensate. Empirically, choosing $m$ in the range 200–400 works generally well across model sizes, matching our heuristic of $m \approx 0.1 d_{\text{out}}$ given typical $d_{\text{out}} \in [3000, 5000]$.

## 6   CONCLUSION

In this work, we introduced *SeedPrints*, a stronger, intrinsic notion of LLM fingerprinting that traces a model's lineage back to its random initialization. We showed that initialization itself leaves a persistent "Galtonian" fingerprint on neural language models. Therefore, untrained models already exhibit stable, seed-dependent token-preference patterns, and these biases persist—statistically and measurably—throughout training. Building on this observation, we proposed a distribution-correlation test over *identity dimensions* that detects shared lineage with calibrated significance. Across LLaMA- and Qwen-style families and a broad suite of realistic deployments (e.g., continued pretraining on divergent corpora, instruction tuning, PEFT, quantization, merging, and distillation), SeedPrints enables "birth-to-lifecycle" verification and remains robust under domain shift, offering a practical tool for provenance and copyright auditing.

**Limitations and Future Directions**    This work aims to uncover fingerprint signals rooted in initialization-stage bias. The proposed method requires access to model outputs (e.g., logits or hidden states) and may not apply when only restricted API access is available. However, we stress that the pursuit of *true fingerprints* for LLMs is still at an early stage: most existing approaches (especially black-box methods) rely on signals that emerge in later training stages or from alignment-driven artifacts and therefore cannot robustly provide protection at the pretrained stage. Therefore, reliable all-stage fingerprints are even more urgently needed. See full discussions in Section A.2.

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

## Appendix Contents

## A    LLM USAGE AND LIMITATION DISCUSSIONS

### A.1    LLM USAGE

We use AI assistants, i.e., ChatGPT and Gemini, for writing and formatting support. Their use covers grammar and style checks, improving clarity of figure and table captions, and other surface-level edits. For programming-related tasks, we occasionally use GitHub Copilot and Claude as coding assistants, e.g., for code auto-completion and debugging hints.

### A.2    LIMITATIONS AND FUTURE DIRECTIONS

Although this work uncovers seed-born, training-persistent biases that can serve as fingerprint signals, our method remains within the white-box paradigm. It may not apply when only restricted API access is available; for instance, when responses are deterministic or safety-filtered. However, we emphasize that the pursuit of *true fingerprints* for LLM is still at a very early stage, even under the white-box setting. Most existing fingerprinting approaches rely on training-induced signatures or alignment behaviors, and thus cannot robustly protect pretrained or early-stage models. In analogy to human fingerprints, which remain stable regardless of age or environment, we believe that a true LLM fingerprint should persist across continued training and deployment contexts rather than reflect incidental training artifacts. Achieving this–particularly under black-box access, where there remains a substantial performance gap compared to white-box methods–remains a challenging open direction and requires substantial future work. We hope this work motivates further research toward establishing principled, robust fingerprinting frameworks for large models.

## B    VALIDATION OF THE EMPIRICAL AND ANALYTICAL NULL DISTRIBUTION

Our hypothesis test in Section 4 evaluates whether two models share lineage by comparing the distribution of their correlation statistics over random inputs against a null distribution. The null distribution models the correlation statistics between two *independent models evaluated on random inputs*.

We consider two approaches for approximating this null distribution: (1) empirical simulation-based null obtained by applying the full pipeline to Gaussian surrogate outputs, and (2) an analytical null derived from the known distribution of Kendall–Tau correlations. In this section, we justify these approximations through three steps: (i) characterizing the true null using independently initialized models, (ii) introducing the Gaussian surrogate as a tractable simulation-based proxy, and (iii) validating the analytical approximation.

### B.1    EMPIRICAL TRUE NULL FROM INDEPENDENTLY INITIALIZED MODELS

The ideal null distribution corresponds to the correlation statistics between two independent models evaluated on random inputs. The cleanest way to instantiate this independence is to compare two independently initialized models (with different random seeds and no training). We estimate this empirical null by evaluating $\sim 2500$ such pairs and computing Kendall–Tau correlations on 10,000 random inputs (as in Section 3). Figure 3 shows that the true null distribution is empirically a Gaussian centered around zero.

### B.2    GAUSSIAN SURROGATE AS A SIMULATION-BASED APPROXIMATION

Directly estimating the null distribution using independently initialized models is computationally expensive, as it requires access to many models and repeated evaluations. To obtain a tractable alternative, we introduce a simulation-based null by using Gaussian surrogates as proxies for model outputs under the null.

This choice is motivated by two considerations. First, approximating neural network outputs with Gaussian distributions is a widely adopted practice in auditing literature, supported by both empirical and theoretical studies (Carlini et al., 2022; Lee et al., 2019). Second, our pipeline depends only on relative ordering and selection operations (e.g., rank correlations and bottom-$m$ selection), and does not rely on semantic structure in the inputs. Moreover, randomly initialized models have

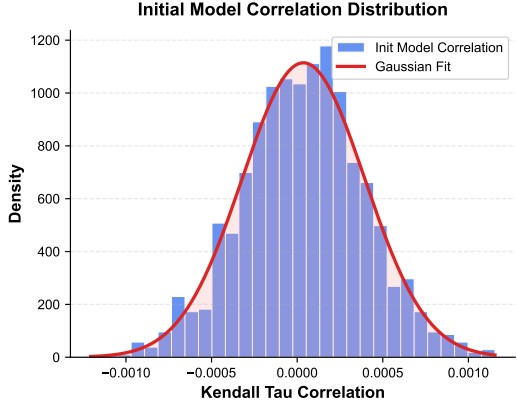

Figure 3: Empirical correlation distribution over $\sim 2500$ pairs of independently initialized models, evaluated on random inputs, closely matches a Gaussian distribution centered at zero.

independent parameter matrices, suggesting that their outputs under the null exhibit no strong structured dependencies. Therefore, sampling independent Gaussian matrices provides a simple and tractable proxy for modeling the null behavior of independently initialized models.

We construct the surrogate null by sampling Gaussian matrices and applying the full pipeline, including identity-dimension selection and correlation computation:

1. Sample two Gaussian matrices of shape $\mathbb{R}^{N \times d_{\text{out}}}$,
2. Extract the most biased dimensions by selecting the bottom-$m$ ranked average dimensions,
3. Take the intersection set $\mathcal{S}$ and compute correlations on it.

This simulation-based construction applies the same selection procedure to Gaussian samples and naturally captures the dependency patterns introduced by the pipeline itself (e.g., those arising from selecting $\mathcal{S}$ across dimensions). As shown in Figure 4 (middle), the resulting distribution matches the empirical null obtained from independently initialized models in terms of its centered Gaussian shape, indicating that the Gaussian surrogate provides an effective simulation-based approximation of the true null.

### B.3 ANALYTICAL APPROXIMATION UNDER WEAK DEPENDENCE

While the Gaussian surrogate captures the behavior of the null distribution under the full pipeline, it still requires repeated sampling and introduces randomness in the estimation process. We therefore also propose the most efficient analytical approximation.

Under the null hypothesis that the two models are independent[2], each Kendall–Tau correlation $\tau_j$ is approximately zero-mean with known variance $\sigma^2$. If correlations across identity dimensions were independent, the central limit theorem would imply

$$\bar{\tau} = \frac{1}{|\mathcal{S}|} \sum_{j \in \mathcal{S}} \tau_j \ \sim \ \mathcal{N}\left(0, \frac{\sigma^2}{|\mathcal{S}|}\right).$$

In practice, softmax normalization introduces dependencies across dimensions. However, Figure 4 shows that the analytical Gaussian closely matches the distribution obtained from the Gaussian surrogate pipeline across different settings, including varying numbers of random inputs $n$ and output dimensions $D$. This suggests that such dependencies are sufficiently weak and do not materially affect the accuracy of the approximation. A plausible explanation is that the use of a relatively high temperature ($T = 10$) smooths the output distribution and reduces coupling between dimensions.

Together, these results justify the use of the analytical Gaussian null in Section 4 as an efficient, deterministic, and accurate approximation, eliminating the need for costly empirical baseline construction.

---

[2]As discussed in Section B.2, although outputs may exhibit weak dependencies due to shared inputs, this approximation remains accurate empirically.

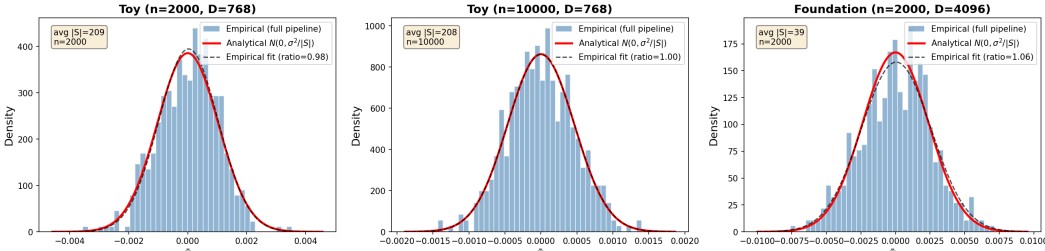

Figure 4: Validation of the analytical Gaussian null. We apply the full pipeline (including identity-dimension selection and correlation aggregation) to Gaussian surrogate outputs and compare the resulting distribution of $\bar{\tau}$ with the analytical Gaussian $\mathcal{N}(0, \sigma^2/|\mathcal{S}|)$. The close match indicates that the analytical approximation accurately captures the null behavior, despite weak dependencies introduced by the pipeline.

**On variance scaling.** The true null obtained from randomly initialized models has slightly smaller variance than the Gaussian null approximation. This mismatch does not affect our hypothesis test results. Our primary evaluation for simulation-based null in Section D.2 relies on the Mann–Whitney U-test, which is rank-based and thus invariant to any positive scaling: for any $c > 0$, $x_i < x_j \Leftrightarrow cx_i < cx_j$. Therefore, all rank statistics and rejection thresholds remain unchanged. For tests that do depend on absolute variance, we have additionally evaluated variance-sensitive statistics such as the $t$-test, whose statistic scales as $t' = t/c$ when variance decreases. Since $c < 1$ in our case, scaling reduces the magnitude of $t$ and makes the test more conservative rather than inflating significance. Empirically, across extensive experiments in Section D.2, we find that the variance scaling does not affect the power of the $t$-test, indicating that the distributional separation between the alternative hypothesis and the null is substantially larger than the scaled variance.

For the analytical null used in the main paper, the variance is explicitly included in the normalization of the $z$-score. Therefore, any scale discrepancy is absorbed in the denominator. Since the analytical variance is slightly larger than the empirical one, the resulting $z$-scores are smaller, making the test conservative and preventing false positives.

## C  THEORETICAL PROPERTIES OF OUR METHODS

Our goal is to identify the output dimensions that the initialized model is least likely to predict under random inputs. These dimensions correspond to directions that the model inherently disfavors due to initialization-induced bias (not dependent on the input distribution), forming the basis of a stable fingerprint signal.

**Setup.** Consider a model $g$ with fixed parameters and random inputs $x_i$ (uniform random tokens or random embeddings as in the main paper). For each output dimension $j \in [d_{\text{out}}]$, $g_j(x)$ is thus a random variable and its population mean is $\bar{g}_j := \mathbb{E}[g_j(x)]$. The population-level directions we aim to capture correspond to the $m$ output dimensions with the smallest expected values, formally defined as

$$\mathcal{M} = \arg\min_{J \subseteq \{1,\ldots,d_{\text{out}}\}, |J|=m} \sum_{j \in J} \bar{g}_j.$$

Let $(\bar{g}_{(1)}, \ldots, \bar{g}_{(d_{\text{out}})})$ denote the sorted means in non-decreasing order, and write

$$\Gamma(m) := \bar{g}_{(m+1)} - \bar{g}_{(m)} > 0$$

for the population "gap" between the $m$-th and $(m+1)$-th smallest means. This gap measures how well-separated the target set $\mathcal{M}$ is from the remaining coordinates. Importantly, for a fixed $m$ and input distribution, $\Gamma(m)$ depends solely on the model's inherent initialization bias. *However, increasing the length of random input sequences amplifies this bias via concentration, making the separation more pronounced at the population level.*

There are two natural empirical strategies to extract such population-level least-likely dimensions:

1. **Sample-mean-based extraction (ours).** Estimate the expectation of each output coordinate by the empirical mean over random inputs $X = \{x_i\}_{i=1}^n$ and select the $m$ coordinates with the smallest value. Following the notation in Section 4, where $g$ denotes the model (either the base model $f$ or the suspect model $f'$):

$$\widehat{g} = \frac{1}{n} \sum_{i=1}^n g(x_i), \qquad \widehat{\mathcal{M}}^{\text{mean}} = \arg \min_{J \subseteq \{1,\dots,d_{\text{out}}\},\, |J|=m} \sum_{j \in J} \widehat{g}_j.$$

2. **Per-sample argmin voting.** For each random input $x_i \in \mathbb{R}^{\ell \times d}$, select the smallest-value coordinate and then take the $m$ most frequent dimensions:

$$j^*(x_i) = \arg \min_{j \in \{1,\dots,d_{\text{out}}\}} g_j(x_i), \qquad \widehat{\mathcal{M}}^{\text{vote}} = \arg \max_{J \subseteq \{1,\dots,d_{\text{out}}\},\, |J|=m} \sum_{i=1}^n \mathbf{1}\{j^*(x_i) \in J\}.$$

Let $p_j := \Pr(j^*(x) = j)$ denote the population probability that index $j$ attains the minimum, and let

$$\mathcal{M}_{\text{vote}} := \arg \max_{J \subseteq \{1,\dots,d_{\text{out}}\},\, |J|=m} \sum_{j \in J} p_j$$

be the population top-$m$ set under voting.

Both approaches are intuitively reasonable. However, as we show below, the sample-mean-based strategy is provably stable, whereas per-sample argmin voting is discontinuous and can flip under arbitrarily small perturbations, making it inherently noise-sensitive.

### C.1   STABILITY ANALYSIS OF MEAN-BASED IDENTITY EXTRACTION VS. PER-SAMPLE ARGMIN VOTING

#### C.1.1   STABILITY OF SAMPLE-MEAN-BASED EXTRACTION

We first show that the sample-mean-based operator, which selects the $m$ smallest coordinates in the empirical mean vector, is stable under small perturbations to model outputs. The perturbations represents realistic variations in outputs from the *same* model, e.g., numerical noise, or evaluation under different precision modes.

Formally, we model perturbed outputs as $g'(x) = g(x) + \eta(x)$, where $\eta(x) \in \mathbb{R}^{d_{\text{out}}}$ represents stochastic deviations. We assume that for each sample $x_i$ and coordinate $j$, the random variables $\eta_j(x_i)$ are independent across samples (not necessarily across coordinates) and $\sigma$-sub-Gaussian with $\mathbb{E}[\eta_j(x_i)] = 0$.[3]

**Lemma C.1** (Stability of sample-mean-based extraction). *Let $\widehat{g}$ and $\widehat{g}'$ be the empirical means over $X = \{x_i\}_{i=1}^n$, i.e., $\widehat{g} = \frac{1}{n} \sum_{i=1}^n g(x_i)$ and $\widehat{g}' = \frac{1}{n} \sum_{i=1}^n g'(x_i) = \widehat{g} + \frac{1}{n} \sum_{i=1}^n \eta(x_i)$. Let $\widehat{\Gamma}(m)$ denote the empirical gap at rank $m$. Then,*

$$\Pr\Big(\widehat{\mathcal{M}}^{\text{mean}}(g') \neq \widehat{\mathcal{M}}^{\text{mean}}(g)\Big) \leq 2d_{\text{out}} \exp\Big(-\frac{n\widehat{\Gamma}(m)^2}{8\sigma^2}\Big).$$

> **Interpretation.** The lemma provides a quantitative upper bound on the probability that the sample-mean-based extractor changes its selected set under perturbations. This instability probability decreases exponentially with the number of averaged samples $n$, due to variance reduction from $\sigma^2$ per sample to $\sigma^2/n$ in the empirical mean.

*Proof.* We condition on the sample $X = \{x_i\}_{i=1}^n$ throughout the proof, so that $\widehat{g}$ and $\widehat{\Gamma}(m)$ are fixed, and all probabilities are taken over the randomness of $\{\eta(x_i)\}_{i=1}^n$ only.

---

[3]This sub-Gaussian assumption is mild: bounded random variables are $\sigma$-sub-Gaussian (Hoeffding's lemma), and common initialization schemes together with bounded activations yield light-tailed behavior in both initialized and trained networks. The Gaussian case $\eta_j(x_i) \sim \mathcal{N}(0, \sigma^2)$ is a special case of this assumption.

For any coordinate $j$, by definition of $\widehat{g}'$, $\widehat{g}'_j - \widehat{g}_j = \frac{1}{n}\sum_{i=1}^{n}\eta_j(x_i)$. Each $\eta_j(x_i)$ is mean-zero and $\sigma$-sub-Gaussian, so their average is mean-zero and $(\sigma/\sqrt{n})$-sub-Gaussian: the variance proxy scales as $1/n$ under averaging. Hence, for any $\varepsilon > 0$, by the standard sub-Gaussian tail bound,

$$\Pr\left(|\widehat{g}'_j - \widehat{g}_j| \geq \varepsilon \mid X\right) \leq 2\exp\left(-\frac{n\varepsilon^2}{2\sigma^2}\right). \tag{1}$$

Now consider the event that *any* coordinate fluctuates by more than $\varepsilon$:

$$\left\{\max_{1 \leq j \leq d_{\text{out}}} |\widehat{g}'_j - \widehat{g}_j| \geq \varepsilon\right\}.$$

Taking a union bound over all $d_{\text{out}}$ coordinates yields

$$\Pr\left(\max_j |\widehat{g}'_j - \widehat{g}_j| \geq \varepsilon \mid X\right) \leq \sum_{j=1}^{d_{\text{out}}}\Pr\left(|\widehat{g}'_j - \widehat{g}_j| \geq \varepsilon \mid X\right) \leq 2d_{\text{out}}\exp\left(-\frac{n\varepsilon^2}{2\sigma^2}\right). \tag{2}$$

We now show that

$$\left\{\widehat{\mathcal{M}}^{\text{mean}}(g') \neq \widehat{\mathcal{M}}^{\text{mean}}(g)\right\} \subseteq \left\{\max_j |\widehat{g}'_j - \widehat{g}_j| \geq \widehat{\Gamma}(m)/2\right\}.$$

Equivalently, if $\max_j |\widehat{g}'_j - \widehat{g}_j| < \widehat{\Gamma}(m)/2$, then the bottom-$m$ identity set remains unchanged.

Let $j_{(1)}, \ldots, j_{(d_{\text{out}})}$ be a permutation of $\{1, \ldots, d_{\text{out}}\}$ such that $\widehat{g}_{j_{(1)}} \leq \widehat{g}_{j_{(2)}} \leq \cdots \leq \widehat{g}_{j_{(d_{\text{out}})}}$. Recall that the empirical gap at rank $m$ is $\widehat{\Gamma}(m) := \widehat{g}_{j_{(m+1)}} - \widehat{g}_{j_{(m)}} > 0$. Denote $\widehat{\mathcal{M}}^{\text{mean}}(g) := \{j_{(1)}, \ldots, j_{(m)}\}$ and $\widehat{\mathcal{M}}_c^{\text{mean}}(g) := \{j_{(m+1)}, \ldots, j_{(d_{\text{out}})}\}$. For any $i \in \widehat{\mathcal{M}}^{\text{mean}}(g)$ and any $\ell \in \widehat{\mathcal{M}}_c^{\text{mean}}(g)$, $\widehat{g}_\ell - \widehat{g}_i \geq \widehat{g}_{j_{(m+1)}} - \widehat{g}_{j_{(m)}} = \widehat{\Gamma}(m)$. With $|\widehat{g}'_i - \widehat{g}_i| < \widehat{\Gamma}(m)/2$ and $|\widehat{g}'_\ell - \widehat{g}_\ell| < \widehat{\Gamma}(m)/2$, we have

$$\widehat{g}'_\ell - \widehat{g}'_i \geq (\widehat{g}_\ell - \tfrac{\widehat{\Gamma}(m)}{2}) - (\widehat{g}_i + \tfrac{\widehat{\Gamma}(m)}{2}) \geq \widehat{\Gamma}(m) - \widehat{\Gamma}(m) = 0.$$

Thus no coordinate in $\widehat{\mathcal{M}}_c^{\text{mean}}(g)$ can become smaller than any coordinate in $\widehat{\mathcal{M}}^{\text{mean}}(g)$ under this deviation bound, and therefore

$$\widehat{\mathcal{M}}^{\text{mean}}(g') = \widehat{\mathcal{M}}^{\text{mean}}(g).$$

Since $\widehat{\Gamma}(m)$ is fixed under conditioning on $X$, plugging $\varepsilon = \widehat{\Gamma}(m)/2$ into (2) yields

$$\Pr\left(\widehat{\mathcal{M}}^{\text{mean}}(g') \neq \widehat{\mathcal{M}}^{\text{mean}}(g)\right) \leq \Pr\left(\max_j |\widehat{g}'_j - \widehat{g}_j| \geq \widehat{\Gamma}(m)/2 \mid X\right) \leq 2d_{\text{out}}\exp\left(-\frac{n\widehat{\Gamma}(m)^2}{8\sigma^2}\right).$$

$\square$

### C.1.2 INSTABILITY OF PER-SAMPLE ARGMIN VOTING

In contrast, per-sample voting applies a discontinuous winner-take-all operator on each input. Thus, arbitrarily small perturbations to outputs can change decisions with non-vanishing probability. We still consider the same stochastic perturbation model $g'(x) = g(x) + \eta(x)$.

**Lemma C.2** (Instability of per-sample argmin voting). *For a fixed input $x$, let*

$$j^*(x; g) := \arg\min_j g_j(x), \qquad \Delta_g(x) := \min_{j \neq j^*(x;g)}\{g_j(x) - g_{j^*(x;g)}(x)\} > 0$$

*denote the argmin and its margin. Then, for any fixed $x$,*

$$\Pr\left(j^*(x; g') \neq j^*(x; g) \mid x\right) \geq \Phi\left(-\frac{\Delta_g(x)}{\sqrt{2}\sigma}\right),$$

*and for $n$ i.i.d. inputs $x_1, \ldots, x_n$,*

$$\Pr\left(\exists i \leq n : j^*(x_i; g') \neq j^*(x_i; g)\right) \geq 1 - (1 - \alpha)^n,$$

*where $\alpha := \mathbb{E}_x\left[\Phi\left(-\frac{\Delta_g(x)}{\sqrt{2}\sigma}\right)\right]$.*

> **Interpretation.** Unlike mean extraction, voting does not benefit from variance reduction: each input incurs a direct hard decision based on noisy outputs. Increasing $n$ only increases the number of opportunities for index-flips.

*Proof.* We first bound the per-input change probability $\Pr\big(j^*(x; g') \neq j^*(x; g) \,\big|\, x\big)$. Fix $x$ and write $j^* := j^*(x; g)$ for brevity. By definition of the margin, $\Delta_g(x) = \min_{j \neq j^*}\big\{g_j(x) - g_{j^*}(x)\big\}$. Let $j^{(2)}$ be an index attaining this minimum, i.e., $g_{j^{(2)}}(x) - g_{j^*}(x) = \Delta_g(x)$.

Under the perturbation $g'(x) = g(x) + \eta(x)$, we have

$$g'_{j^{(2)}}(x) - g'_{j^*}(x) = \big[g_{j^{(2)}}(x) - g_{j^*}(x)\big] + \big[\eta_{j^{(2)}}(x) - \eta_{j^*}(x)\big] = \Delta_g(x) + \big[\eta_{j^{(2)}}(x) - \eta_{j^*}(x)\big].$$

Since $\eta_{j^{(2)}}(x)$ and $\eta_{j^*}(x)$ are independent $\mathcal{N}(0, \sigma^2)$ variables, their difference is Gaussian with

$$\eta_{j^{(2)}}(x) - \eta_{j^*}(x) \sim \mathcal{N}(0, 2\sigma^2).$$

The event that the argmin changes, $j^*(x; g') \neq j^*(x; g)$, certainly occurs whenever $g'_{j^{(2)}}(x) \leq g'_{j^*}(x)$. Hence,

$$\Pr\big(j^*(x; g') \neq j^*(x; g) \,\big|\, x\big) \;\geq\; \Pr\big(g'_{j^{(2)}}(x) \leq g'_{j^*}(x) \,\big|\, x\big).$$

Using the expression above,

$$\Pr\big(g'_{j^{(2)}}(x) \leq g'_{j^*}(x) \,\big|\, x\big) = \Pr\big(g'_{j^{(2)}}(x) - g'_{j^*}(x) \leq 0 \,\big|\, x\big) = \Pr\big(\Delta_g(x) + [\eta_{j^{(2)}}(x) - \eta_{j^*}(x)] \leq 0 \,\big|\, x\big).$$

Let $Z \sim \mathcal{N}(0, 1)$. Since $\eta_{j^{(2)}}(x) - \eta_{j^*}(x) \overset{d}{=} \sqrt{2}\sigma Z$, we obtain

$$\Pr\big(\Delta_g(x) + [\eta_{j^{(2)}}(x) - \eta_{j^*}(x)] \leq 0 \,\big|\, x\big) = \Pr\Big(Z \leq -\frac{\Delta_g(x)}{\sqrt{2}\sigma}\Big) = \Phi\Big(-\frac{\Delta_g(x)}{\sqrt{2}\sigma}\Big).$$

Therefore, $\Pr\big(j^*(x; g') \neq j^*(x; g) \,\big|\, x\big) \geq \Phi\Big(-\frac{\Delta_g(x)}{\sqrt{2}\sigma}\Big)$.

Now consider $n$ i.i.d. inputs $x_1, \ldots, x_n$. Conditional on the inputs, the events

$$E_i := \{j^*(x_i; g') \neq j^*(x_i; g)\} \qquad E_i^c := \{j^*(x_i; g') = j^*(x_i; g)\}$$

are independent across $i$, because the perturbations $\eta(x_i)$ are independent across $i$. Moreover,

$$\Pr(E_i \,|\, x_i) = \Pr\big(j^*(x_i; g') \neq j^*(x_i; g) \,\big|\, x\big) \geq \Phi\Big(-\frac{\Delta_g(x_i)}{\sqrt{2}\sigma}\Big) \geq \Phi\Big(-\frac{\Delta_g(x_i)}{\sqrt{2}\sigma}\Big).$$

Define $q(x_i) := \Pr\big(j^*(x_i; g') \neq j^*(x_i; g) \big| x\big)$, and $\alpha := \mathbb{E}_x\Big[\Phi\Big(-\frac{\Delta_g(x)}{\sqrt{2}\sigma}\Big)\Big]$. Taking expectations over the randomness of $x_i$, we obtain $\mathbb{E}\big[\Pr(E_i \,|\, x_i)\big] = \alpha$. Therefore, by independence across $i$,

$$\Pr\big(E_1^c \cap \cdots \cap E_n^c\big) = \mathbb{E}\Big[\prod_{i=1}^n (1 - q(x_i))\Big] \;\leq\; \mathbb{E}\Big[\prod_{i=1}^n \big(1 - \Phi(-\Delta_g(x_i)/(\sqrt{2}\sigma))\big)\Big] \;\leq\; (1 - \alpha)^n,$$

where the last inequality uses i.i.d. inputs and Jensen's inequality applied to the convex function $u \mapsto \log(1 - u)$ on $u \in [0, 1)$. Therefore,

$$\Pr\Big(\exists i \leq n : j^*(x_i; g') \neq j^*(x_i; g)\Big) = 1 - \Pr(E_1^c \cap \cdots \cap E_n^c) \;\geq\; 1 - (1 - \alpha)^n.$$

Finally, note that the empirical voting set $\widehat{\mathcal{M}}^{\text{vote}}$ is determined by the empirical counts of the dimensions $\{j^*(x_i; g)\}_{i=1}^n$. If the empirical gap between the $m$-th and $(m{+}1)$-th most frequent dimensions is of order $O(1/n)$ (i.e., only a constant number of votes), then a change in $j^*(x_i)$ for a single input $x_i$ is sufficient to swap the ordering across this boundary and thus alter the top-$m$ set. Combining this observation with the lower bound $1 - (1 - \alpha)^n$ shows that, in such regimes, the probability that $\widehat{\mathcal{M}}^{\text{vote}}(g') \neq \widehat{\mathcal{M}}^{\text{vote}}(g)$ is bounded away from zero uniformly in $n$, i.e., per-sample argmin voting does not enjoy variance reduction with more samples. $\square$

## C.2  Influence of Hyperparameters

We now analyze how many random inputs are required for the empirical mean estimator to recover the true bias. We reuse notation from **Setup**: each output coordinate $g_j(x)$ is a random variable under random inputs $x \sim \mathcal{D}$, with population mean $\bar{g}_j := \mathbb{E}[g_j(x)]$. $(\bar{g}_{(1)}, \ldots, \bar{g}_{(d_{\text{out}})})$ represents the sorted means in non-decreasing order, and $\gamma := \bar{g}_{(m+1)} - \bar{g}_{(m)} > 0$ is the separation margin. We estimate $\bar{g}$ using $n$ i.i.d. random inputs:

$$\widehat{g}_j := \frac{1}{n} \sum_{i=1}^{n} g_j(x_i), \qquad \widehat{g} = (\widehat{g}_1, \ldots, \widehat{g}_{d_{\text{out}}}).$$

The target dimensions set is $\mathcal{M}$ with size $m$ and our empirical estimation gives $\widehat{\mathcal{M}}$.

**Lemma C.3** (Sufficient condition for recovering the true set). *If $\|\widehat{g} - \bar{g}\|_\infty \leq \varepsilon$ and $2\varepsilon < \gamma$, then the empirical bottom-$m$ set equals the true set: $\widehat{\mathcal{M}} = \mathcal{M}$.*

*Proof.* For any $j \in \mathcal{M}$ and $j' \notin \mathcal{M}$, we have $\bar{g}_j \leq \bar{g}_{(m)}$ and $\bar{g}_{j'} \geq \bar{g}_{(m+1)}$. The $\ell_\infty$ bound gives

$$\widehat{g}_j \leq \bar{g}_{(m)} + \varepsilon, \qquad \widehat{g}_{j'} \geq \bar{g}_{(m+1)} - \varepsilon.$$

Thus,

$$\widehat{g}_j - \widehat{g}_{j'} \leq -(\gamma - 2\varepsilon) < 0,$$

so no outside index can enter nor inside index leave the bottom-$m$ set. □

We next interpret this recovery condition in terms of hyperparameters. We again adopt the same sub-Gaussian assumption as in Section C.1. This assumption is mild: by Hoeffding's inequality, any bounded random variable is $\sigma$-sub-Gaussian with $\sigma$ proportional to its range. In our setting, each $g_j(x)$ corresponds to a logit or final hidden activation, which are typically bounded or light-tailed in both initialized networks (due to controlled parameter initialization) and trained networks.

**Corollary C.4** (Sample complexity under sub-Gaussian coordinates). *Suppose each $g_j(x)$ is $\sigma$-sub-Gaussian. Then with probability at least $1 - \delta$,*

$$\|\widehat{g} - \bar{g}\|_\infty \leq \sigma \sqrt{\frac{2 \log(2d_{out}/\delta)}{n}}.$$

*Consequently, if*

$$n \geq \frac{8\sigma^2}{\gamma^2} \log\left(\frac{2d_{out}}{\delta}\right),$$

*then $\widehat{\mathcal{M}} = \mathcal{M}$ with probability at least $1 - \delta$.*

*Proof.* For fixed $j$, sub-Gaussian concentration implies

$$\Pr(|\widehat{g}_j - \bar{g}_j| > \varepsilon) \leq 2 \exp(-n\varepsilon^2/2\sigma^2).$$

A union bound over all $d_{\text{out}}$ coordinates yields

$$\Pr\left(\|\widehat{g} - \bar{g}\|_\infty > \varepsilon\right) \leq \sum_{j=1}^{d_{\text{out}}} \Pr(|\widehat{g}_j - \bar{g}_j| > \varepsilon) \leq 2d_{\text{out}} \exp\left(-\frac{n\varepsilon^2}{2\sigma^2}\right).$$

Set the right-hand side to $\delta$ and solve for $\varepsilon$ to obtain, with probability at least $1 - \delta$,

$$\|\widehat{g} - \bar{g}\|_\infty \leq \sigma \sqrt{\frac{2 \log(2d_{\text{out}}/\delta)}{n}}.$$

Now choose

$$\varepsilon := \sigma \sqrt{\frac{2 \log(2d_{\text{out}}/\delta)}{n}}.$$

If $n \geq \frac{8\sigma^2}{\gamma^2} \log\left(\frac{2d_{\text{out}}}{\delta}\right)$ (i.e., if $2\varepsilon < \gamma$), then Lemma C.3 applies and implies $\widehat{\mathcal{M}} = \mathcal{M}$ with probability at least $1 - \delta$. □

Table 9: Fingerprint behavior under different random initializations across model families.

(a) LLaMA-style models

| Seed Pair | Logits Output | | Hidden State | |
|---|---|---|---|---|
| | $t$-test | $U$-test | $t$-test | $U$-test |
| $s_{42}$ vs. $s_{2000}$ | 0.404 | 0.456 | 0.357 | 0.532 |
| $s_{123}$ vs. $s_{42}$ | 0.214 | 0.295 | 0.678 | 0.565 |
| $s_{1000}$ vs. $s_{123}$ | 0.219 | 0.246 | 0.363 | 0.335 |
| $s_{2000}$ vs. $s_{1000}$ | 0.282 | 0.291 | 0.434 | 0.481 |

(b) Qwen-style models

| Seed Pair | Logits Output | | Hidden State | |
|---|---|---|---|---|
| | $t$-test | $U$-test | $t$-test | $U$-test |
| $s_{123}$ vs. $s_{1000}$ | 0.741 | 0.727 | 0.094 | 0.074 |
| $s_{1000}$ vs. $s_{123}$ | 0.954 | 0.971 | 0.125 | 0.094 |
| $s_{42}$ vs. $s_{2000}$ | 0.273 | 0.360 | 0.451 | 0.529 |
| $s_{2000}$ vs. $s_{42}$ | 0.215 | 0.206 | 0.230 | 0.295 |

Table 10: (LLaMA-style models) Trained models share the same fingerprint behaviors as their initialization ($p$-value $< 0.01$).

| Model Pair | Logits Output | | Hidden State | | Baselines | | | |
|---|---|---|---|---|---|---|---|---|
| | $t$-test | $u$-test | $t$-test | $u$-test | Intrinsic | REEF | PCS | ICS |
| $s_{42}^{init}$ vs. $s_{42}^{base}$ | 3.33e-3✓ | 1.02e-3✓ | 2.20e-8✓ | 6.28e-8✓ | -0.021✗ | 0.375✗ | 0.580✗ | 0.196✗ |
| $s_{123}^{init}$ vs. $s_{123}^{base}$ | 2.06e-3✓ | 7.33e-3✓ | 7.09e-6✓ | 1.37e-5✓ | 0.149✗ | 0.369✗ | 0.581✗ | 0.188✗ |
| $s_{1000}^{init}$ vs. $s_{1000}^{base}$ | 2.44e-3✓ | 4.14e-3✓ | 5.58e-4✓ | 2.81e-3✓ | -0.252✗ | 0.381✗ | 0.581✗ | 0.188✗ |
| $s_{2000}^{init}$ vs. $s_{2000}^{base}$ | 5.63e-3✓ | 6.76e-3✓ | 4.00e-10✓ | 1.27e-9✓ | -0.337✗ | 0.331✗ | 0.581✗ | 0.188✗ |

Therefore, accurate recovery requires $n = \Omega\left(\frac{\log d_{\text{out}}}{\gamma^2}\right)$, meaning the number of random queries grows logarithmically with the output dimension, but scales inversely with the squared separation margin $\gamma$. The margin $\gamma$ itself has two contributing factors:

- **Intrinsic bias strength.** Longer random input sequences amplify initialization-induced bias via concentration, increasing the population separation between coordinates.

- **Choice of $m$.** Different choices of $m$ correspond to different parts of the ranked spectrum $(\bar{g}_{(1)}, \ldots, \bar{g}_{(d_{\text{out}})})$, thus yielding different effective margins in practice.

The former can be controlled by the input length $l$, while the later is selected empirically based on the observed gaps in the mean spectrum.

## D    MORE EXPERIMENT DETAILS

### D.1    IMPLEMENTATION DETAILS AND LICENSES

We train all models with the Hugging Face Transformers Trainer (Wolf et al., 2020), using Accelerate (Gugger et al., 2022) for distributed runs. All open-source models are loaded from their official Hugging Face releases and used under their original licenses: Llama models under the Meta Llama Community License, and other models under Apache-2.0. All datasets are downloaded via the Hugging Face Datasets library (the library is Apache-2.0); dataset content follows each dataset's stated license.

### D.2    COMPLEMENTARY SEEDPRINTS RESULTS USING SIMULATION-BASED NULL

This section contains the complementary results for Section 5. We report results using the empirical simulation-based null in Algorithm 1 with both *logits* and *hidden state* outputs, and we test with both the one-sided $t$-test ($t$-test) and Mann–Whitney $U$ test ($U$ test).

#### D.2.1    BIRTH-TO-LIFECYCLE "BIOMETRIC" FINGERPRINTING

We train 12-layer, 12-head LLaMA-style models (Touvron et al., 2023) with RoPE (Su et al., 2021) and Qwen-style models (Team, 2024) from scratch. Because the simulation-based random baseline is stochastic, we report $p$-values averaged over 10 independent trials and adopt a significance level of $\alpha = 0.01$. Importantly, the absolute magnitude of extremely small $p$-values is not meaningful:

Table 11: (Qwen-style models) Trained models share the same fingerprint behaviors as their initialization models ($p$-value $< 0.01$).

| Model Pair | Logits Output | | Hidden State | |
|---|---|---|---|---|
| | $t$-test | $U$-test | $t$-test | $U$-test |
| $s_{123}^{init}$ vs. $s_{123}^{base}$ | 2.38e-03 | 1.68e-03 | 7.36e-15 | 3.38e-13 |
| $s_{1000}^{init}$ vs. $s_{1000}^{base}$ | 1.35e-04 | 4.84e-05 | 4.41e-13 | 2.01e-11 |
| $s_{42}^{init}$ vs. $s_{42}^{base}$ | 1.39e-03 | 1.46e-03 | 1.06e-24 | 2.05e-19 |
| $s_{2000}^{init}$ vs. $s_{2000}^{base}$ | 1.80e-03 | 1.28e-03 | 4.87e-24 | 1.92e-20 |

Table 12: Fingerprint persistence under continual training on diverse datasets (base model: seed 1000, corpus `openwebtext`). $U$-test refers to the Mann–Whitney $U$ test.

(a) LLaMA-style models

| Setting | Ours (logits) | | Ours (hidden) | | Baselines | | | |
|---|---|---|---|---|---|---|---|---|
| Continual corpus (seed) | $t$-test | $U$-test | $t$-test | $U$-test | Intrinsic | REEF | PCS | ICS |
| `TinyStories` (1000) | 0 ✓ | 0 ✓ | 0 ✓ | 7.77e-89 ✓ | 1.000 ✓ | 0.759 ✗ | 0.999 ✓ | 0.996 ✓ |
| `TinyStories` (123) | 1.000 ✓ | 1.00 ✓ | 0.943 ✓ | 0.902 ✓ | 0.950 ✗ | 0.658 ✓ | 0.332 ✓ | 0.012 ✓ |
| `the_stack` (1000) | 0 ✓ | 1.73e-287 ✓ | 0 ✓ | 3.09e-69 ✓ | 0.489 ✗ | 0.557 ✗ | 0.585 ✗ | 0.123 ✗ |
| `the_stack` (123) | 0.616 ✓ | 0.479 ✓ | 0.732 ✓ | 0.831 ✓ | 0.445 ✓ | 0.580 ✓ | 0.301 ✓ | 0.026 ✓ |

(b) Qwen-style models

| Setting | Ours (logits) | | Ours (hidden) | | Baselines | | | |
|---|---|---|---|---|---|---|---|---|
| Continual corpus (seed) | $t$-test | $U$-test | $t$-test | $U$-test | Intrinsic | REEF | PCS | ICS |
| `TinyStories` (1000) | $5.36e-09$ ✓ | $1.92e-07$ ✓ | $8.49e-214$ ✓ | $5.09e-71$ ✓ | 1.000 ✓ | 0.957 ✓ | 0.999 ✓ | 0.996 ✓ |
| `TinyStories` (123) | 0.434 ✓ | 0.433 ✓ | 0.256 ✓ | 0.065 ✓ | 0.913 ✗ | 0.199 ✓ | 0.328 ✓ | 0.039 ✓ |
| `the_stack` (1000) | 0 ✓ | $4.16e-237$ ✓ | $1.16e-211$ ✓ | $2.30e-76$ ✓ | 0.999 ✓ | 0.313 ✗ | 0.995 ✓ | 0.976 ✓ |
| `the_stack` (123) | 0.999 ✓ | 0.993 ✓ | 0.610 ✓ | 0.491 ✓ | 0.916 ✗ | 0.255 ✓ | 0.328 ✓ | 0.038 ✓ |

once $p$ falls below numerical and sampling noise (e.g., $< 10^{-20}$), values like $10^{-260}$ should not be interpreted as stronger evidence than $10^{-20}$—both decisively reject the null.

**Different initialization seeds produce distinct fingerprints** Table 9a and Table 9b report $p$-values from our correlation tests between pairs of models initialized with different random seeds (42, 123, 1000, and 2000) under the $t$-test and $U$-test, for Llama-style model and Qwen-style model, respectively. All $p$-values are consistently $> 0.01$, indicating that our method reliably distinguishes models trained from different seeds. This shows that distinct seeds yield distinct fingerprint behaviors, allowing models to be separated "at birth."

**Training preserves the initialization fingerprint.** Table 10 (LLaMA-style) and Table 11 (Qwen-style) compares each initialization model $s^{init}$ with its descendant $s^{base}$ trained on the OpenWebText dataset (Gokaslan et al., 2019) ($\approx$10B tokens). Across all seed–model pairs, $p$-values are consistently $< 0.01$, indicating that their bias profiles remain strongly correlated and thus share a common lineage. In short, the trained model inherits the same fingerprint as its initialization; training does not erase the initialization fingerprint. We also evaluate baseline methods in Table 10; without exception, they fail to distinguish across seeds, which in turn suggests their separability stems from training-induced artifacts rather than initialization.

**Identical data and order do not make fingerprints converge** In Table 13a (LLaMA-style models) and Table 13b (Qwen-style models), all four "suspicious" models $s_i^{base}$ for $i \in \{42, 123, 100, 2000\}$ are trained on *exactly the same corpus (OpenWebText) and in the same data order* (we fix the training seed to lock the data order); the only difference lies in their initialization seeds $i$. We aim to test whether fingerprint behavior would be erased or confounded by identical data and order. Across all

Table 13: The same dataset and training order do not shape fingerprint behaviors to be identical across different initializations.

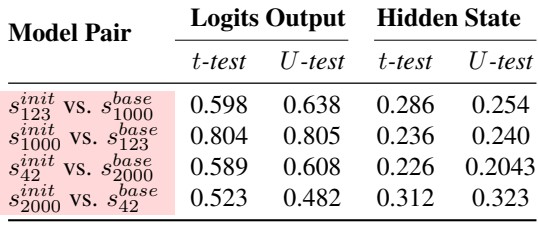

(a) LLaMA-style models

(b) Qwen-style models

| Model Pair | Logits Output | | Hidden State | | Model Pair | Logits Output | | Hidden State | |
|---|---|---|---|---|---|---|---|---|---|
| | $t$-test | $U$-test | $t$-test | $U$-test | | $t$-test | $U$-test | $t$-test | $U$-test |
| $s_{123}^{init}$ vs. $s_{1000}^{base}$ | 0.484 | 0.500 | 0.385 | 0.486 | $s_{123}^{init}$ vs. $s_{1000}^{base}$ | 0.598 | 0.638 | 0.286 | 0.254 |
| $s_{1000}^{init}$ vs. $s_{2000}^{base}$ | 0.946 | 0.956 | 0.135 | 0.096 | $s_{1000}^{init}$ vs. $s_{123}^{base}$ | 0.804 | 0.805 | 0.236 | 0.240 |
| $s_{42}^{init}$ vs. $s_{123}^{base}$ | 0.598 | 0.589 | 0.426 | 0.337 | $s_{42}^{init}$ vs. $s_{2000}^{base}$ | 0.589 | 0.608 | 0.226 | 0.2043 |
| $s_{2000}^{init}$ vs. $s_{42}^{base}$ | 0.756 | 0.781 | 0.388 | 0.287 | $s_{2000}^{init}$ vs. $s_{42}^{base}$ | 0.523 | 0.482 | 0.312 | 0.323 |

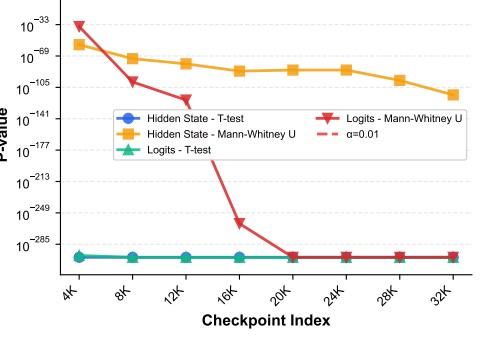

(a) LLaMA-style models

(b) Qwen-style models

Figure 5: Fingerprint verifies lineage at every checkpoint ($p$-values $< 0.01$) across model families.

cross-seed pairs, $p$-values remain consistently $> 0.01$, in sharp contrast to the near-zero values in Table 10. That is, fingerprints remain seed-specific even under identical data and curriculum.

**Continual training on diverse datasets does not confound the fingerprint** The purpose of our earlier experiments is solely to demonstrate the strengths of our SeedPrints: it can act as a biometric fingerprint. From a copyright perspective, the inability of prior works to distinguish models with different seeds is not a weakness, since initialization seeds have no clear copyright status. The real fragility is that their attribution can be easily misled by *data distribution*, ***failing to recognize lineage when the training distribution shifts substantially during continued training.***

In Table 12a and Table 12b, we continue training a base model (seed 1000, pretrained on OpenWeb-Text (Gokaslan et al., 2019)) on two very different datasets: TinyStories (Eldan & Li, 2023) (synthetic children's stories) and The Stack (Kocetkov et al., 2022) (permissively licensed GitHub code). We compare (i) true descendants trained from the base, versus (ii) *distractors* derived from a different base model (initialized with seed 123 and trained with a different data order on OpenWebText), then continued training on the *same* corpus. The question is whether attribution methods can identify which descendant truly shares lineage with the base.

We find that prior baselines all fail under the code setting (The Stack), misclassifying true descendants as distractors. This indicates that they largely track domain similarity rather than lineage identity: TinyStories is closer in distribution to the pretraining corpus (OpenWebText), while The Stack diverges sharply; such a large distribution shift can easily bypass detection. In contrast, our method correctly attributes lineage across both corpora. Hence, our fingerprint is not a proxy for data distribution: it survives substantial domain shift and persists beyond the initial pretraining stage.

### D.2.2 ALL-STAGE VERIFIABLE FINGERPRINTS

**Controlled pretraining trajectory.** We first conduct a controlled pretraining experiment on Open-WebText, where we know the initialization seed. Each intermediate checkpoint is treated as the base, and we test whether our method can reliably identify its offspring along the same training trajectory. All variants consistently recognize the suspect model as belonging to the same lineage,

Table 14: Cross-size similarity between Qwen-2.5-14B (target) and other Qwen-2.5 models.

|  | Qwen-2.5-0.5B | Qwen-2.5-1.5B | Qwen-2.5-3B | Qwen-2.5-7B |
|---|---|---|---|---|
| SeedPrint ($\uparrow$) | 0.7184 | 0.9707 | 0.3898 | 0.5429 |
| REEF ($\downarrow$) | 0.8250 | 0.8399 | 0.8833 | 0.8461 |
| HUREF ($\downarrow$) | $-6.37\times10^{-5}$ | $-2.84\times10^{-5}$ | $-8.70\times10^{-6}$ | $-6.08\times10^{-6}$ |

with $p$-values remaining below the 0.01 threshold (Figure 5a for LLaMA-style models and Figure 5b for Qwen-style models). This confirms the stability of SeedPrints under continuous optimization.

### D.3    CROSS-SIZE FINGERPRINT WITHIN THE QWEN-2.5 SERIES

To further examine whether SeedPrints "collapse" within a model family when we vary the parameter size, we report cross-size similarity between Qwen-2.5-14B (target) and smaller models from the same Qwen-2.5 family. Table 14 shows the results for SeedPrint ($\uparrow$) and two baselines, REEF ($\downarrow$) and HUREF ($\downarrow$).

To probe whether initialization-born fingerprints "collapse" to a family-level signature when scaling model size, we conduct a cross-size study within the Qwen-2.5 series. We treat Qwen-2.5-14B as the protected target model and consider Qwen-2.5-{0.5B, 1.5B, 3B, 7B} as suspect models that share the same architectural family but are not trained as descendants of Qwen-2.5-14B. As discussed in the main paper, SeedPrints outputs a $p$-value for the hypothesis test "the suspect shares initialization-born fingerprints with the target", while REEF and HUREF are similarity-based baselines that directly output continuous scores and require the user to choose a threshold.

Several observations are noteworthy. First, all SeedPrints $p$-values are comfortably above a typical significance level (e.g., $\alpha=0.01$). Under our hypothesis-testing view, this means that none of the smaller Qwen-2.5 models is flagged as sharing the same lineage as Qwen-2.5-14B. This behavior is desirable in our setting: even though these models belong to the same family and reuse a similar architecture design, the fingerprinting test does not collapse to a single family-level signature that would spuriously conflate distinct pre-training runs.

The cross-size $p$-values occupy different regions of the $[0, 1]$ range rather than concentrating around a single value. We do *not* interpret these magnitudes as a continuous "similarity score"—their role is to support a calibrated yes/no decision at a fixed significance level. Nonetheless, the spread indicates that our test statistic remains sensitive to the concrete combination of architecture and initialization seed, instead of degenerating into a generic pattern shared by all Qwen-2.5 variants.

Finally, the similarity-based baselines behave differently. REEF produces scores in a relatively narrow band (0.8250–0.8833) across all sizes, while HUREF outputs values that are numerically very close to zero (on the order of $10^{-5}$). Since these methods do not provide a built-in decision rule, operators must manually choose thresholds, and it is unclear how to robustly separate cross-size variants from genuinely lineage-related models using a single cutoff. In contrast, SeedPrints directly yields a statistical decision at a prescribed significance level, providing a clearer and more operational answer to the question "do these two models plausibly share a training lineage?" within a model family.

### E    MODEL CATALOG AND DECODING SETTINGS

Our evaluation covers **58** models in total. Below we document the decoding configuration used throughout all runs and enumerate the model catalog grouped by family. For each derived model we also indicate the post-training transformation (*finetune*, *adapter*, *merge*, *quantization*, *distillation*). "PT" and "IT" refer to *pre-trained* and *instruction-tuned* source models, respectively.

| Parameter | Value |
|---|---|
| max_new_tokens | 512 |
| temperature | 0.7 |
| top_p | 0.9 |
| top_k | 50 |
| do_sample | true |
| max_input_length | 512 |

## E.1 MODEL LIST BY FAMILY

### E.1.1 QWEN-2.5-7B

| Variant | HuggingFace repo | Type |
|---|---|---|
| PT | Qwen/Qwen2.5-7B | — |
| IT | Qwen/Qwen2.5-7B-Instruct | — |
| Derived | Qwen/Qwen2.5-Math-7B | finetune |
| Derived | Qwen/Qwen2.5-Coder-7B-Instruct | finetune |
| Derived | WangCa/Qwen2.5-7B-Medicine | finetune |
| Derived | huihui-ai/Qwen2.5-7B-Instruct-abliterated-v2 | finetune |
| Derived | Locutusque/StockQwen-2.5-7B | merge |
| Derived | bunnycore/QevaCoT-7B-Stock | merge |
| Derived | fangcaotank/task-10-Qwen-Qwen2.5-7B-Instruct | adapter |
| Derived | SeeFlock/task-12-Qwen-Qwen2.5-7B-Instruct | adapter |
| Derived | Qwen/Qwen2.5-7B-Instruct-GPTQ-Int4 | quantization |
| Derived | Qwen/Qwen2.5-7B-Instruct-GPTQ-Int8 | quantization |
| Derived | Lansechen/Qwen2.5-7B-Open-R1-Distill | distillation |

### E.1.2 QWEN2.5-14B

| Variant | HuggingFace repo | Type |
|---|---|---|
| PT | Qwen/Qwen2.5-14B | — |
| IT | Qwen/Qwen2.5-14B-Instruct | — |
| Derived | Qwen/Qwen2.5-Coder-14B | finetune |
| Derived | oxyapi/oxy-1-small | finetune |
| Derived | v000000/Qwen2.5-14B-Gutenberg-Instruct-Slerpeno | merge |
| Derived | ToastyPigeon/qwen-story-test-qlora | adapter |
| Derived | Qwen/Qwen2.5-14B-Instruct-GPTQ-Int4 | quantization |
| Derived | deepseek-ai/DeepSeek-R1-Distill-Qwen-14B | distillation |

### E.1.3 LLAMA-3.1-8B

| Variant | HuggingFace repo | Type |
|---|---|---|
| PT | meta-llama/Llama-3.1-8B | — |
| IT | meta-llama/Llama-3.1-8B-Instruct | — |
| Derived | ValiantLabs/Llama3.1-8B-Fireplace2 | finetune |
| Derived | RedHatAI/Llama-3.1-8B-tldr | finetune |
| Derived | proxectonos/Llama-3.1-Carballo | finetune |
| Derived | mlabonne/Meta-Llama-3.1-8B-Instruct-abliterated | finetune |
| Derived | gaverfraxz/Meta-Llama-3.1-8B-Instruct-HalfAbliterated-TIES | merge |
| Derived | Xiaojian9992024/Llama3.1-8B-ExtraMix | merge |
| Derived | LlamaFactoryAI/Llama-3.1-8B-Instruct-cv-job-description-matching | adapter |
| Derived | chchen/Llama-3.1-8B-Instruct-PsyCourse-fold7 | adapter |
| Derived | iqbalamo93/Meta-Llama-3.1-8B-Instruct-GPTQ-Q_8 | quantization |
| Derived | DaraV/LLaMA-3.1-8B-Instruct-INT4-GPTQ | quantization |
| Derived | asas-ai/Llama-3.1-8B-Instruct-Open-R1-Distill | distillation |

### E.1.4 MISTRAL-7B-V0.3

| Variant | HuggingFace repo | Type |
|---------|------------------|------|
| PT | mistralai/Mistral-7B-v0.3 | — |
| IT | mistralai/Mistral-7B-Instruct-v0.3 | — |
| Derived | KurmaAI/AQUA-7B | finetune |
| Derived | openfoodfacts/spellcheck-mistral-7b | finetune |
| Derived | grimjim/Mistral-7B-Instruct-demi-merge-v0.3-7B | merge |
| Derived | chaymaemerhrioui/mistral-Brain_Model_ACC_Trainer | adapter |
| Derived | RedHatAI/Mistral-7B-Instruct-v0.3-GPTQ-4bit | quantization |
| Derived | eganwo/mistral7b-distilled-from-deepseek-r1-qwen32b | distillation |

### E.1.5 GEMMA-2-2B

| Variant | HuggingFace repo | Type |
|---------|------------------|------|
| PT | google/gemma-2-2b | — |
| IT | google/gemma-2-2b-it | — |
| Derived | rinna/gemma-2-baku-2b | finetune |
| Derived | anakin87/gemma-2-2b-neogenesis-ita | finetune |
| Derived | vonjack/gemma2-2b-merged | merge |
| Derived | google-cloud-partnership/gemma-2-2b-it-lora-sql | adapter |
| Derived | qilowoq/gemma-2-2B-it-4Bit-GPTQ | quantization |
| Derived | Syed-Hasan-8503/Gemma-2-2b-it-distilled | distillation |

### E.1.6 LLAMA-2-7B

| Variant | HuggingFace repo | Type |
|---------|------------------|------|
| PT | meta-llama/Llama-2-7b-hf | — |
| IT | meta-llama/Llama-2-7b-chat-hf | — |
| Derived | allenai/tulu-2-7b | finetune |
| Derived | QIAIUNCC/EYE-Llama_qa | finetune |
| Derived | DevQuasar/coma-7B-v0.1 | merge |
| Derived | Ammar-1/llama2-Better-Tune | adapter |
| Derived | TheBloke/Llama-2-7B-Chat-GPTQ | quantization |
| Derived | cygu/llama-2-7b-logit-watermark-distill-kgw-k1-gamma0.25-delta2 | distillation |

### E.1.7 TINYLLAMA-1.1B

| Variant | HuggingFace repo | Type |
|---------|------------------|------|
| PT/IT | TinyLlama/TinyLlama-1.1B-Chat-v1.0 | — |
| Derived | alexredna/TinyLlama-1.1B-Chat-v1.0-reasoning-v2 | finetune |
| Derived | Edentns/DataVortexTL-1.1B-v0.1 | finetune |
| Derived | appvoid/dot-v2.7 | merge |
| Derived | barissglc/tinyllama-tarot-v1 | adapter |
| Derived | TheBloke/TinyLlama-1.1B-Chat-v1.0-GPTQ | quantization |
| Derived | anudaw/distilled-code-llama | distillation |

**Notes.** "Type" denotes the post-training transformation relative to the PT/IT source model: *finetune* includes supervised/preference optimization variants; *adapter* includes LoRA/QLoRA-style modules; *merge* includes model soups and TIES-style merges; *quantization* includes GPTQ/INTx variants; *distillation* includes student models distilled from reasoning-augmented teachers.

### E.2 SAMPLE RESULTS

We present partial results from one LeafBench evaluation run in Table 15 ($u$-test with simulation-based null). Reported metrics are $1 - p$, used as similarity scores. We adopt a significance level of $0.01$ for $p$-values; empirically, LeafBench identifies an optimal decision threshold of $0.9920$, further supporting the reliability of our method.

Crucially, while individual $p$-values should not be interpreted as linear similarity measures, their distribution across model pairs is informative: same-lineage models yield much lower $p$-values, while unrelated models remain near uniform.

| Model | Qwen-2.5-7B | Qwen-2.5-7B-Instruct | Qwen2.5-14B | Qwen2.5-14B-Instruct |
|---|---|---|---|---|
| Qwen-2.5-7B | 1.000 | 1.000 | 0.079 | 0.000 |
| Qwen-2.5-7B-Instruct | 1.000 | 1.000 | 0.813 | 0.536 |
| Qwen2.5-7B-Math | 1.000 | 0.994 | 0.047 | 0.987 |
| Qwen2.5-7B-Coder | 0.996 | 0.999 | 0.177 | 0.002 |
| Qwen2.5-7B-Instruct-Medicine | 1.000 | 1.000 | 0.576 | 0.369 |
| Qwen2.5-7B-Instruct-Abilierated | 1.000 | 1.000 | 0.276 | 0.886 |
| Qwen2.5-7B-Stock | 1.000 | 1.000 | 0.939 | 0.225 |
| QevaCoT-7B | 1.000 | 1.000 | 0.199 | 0.119 |
| Qwen2.5-7B-Instruct-Task-10 | 1.000 | 1.000 | 0.672 | 0.090 |
| Qwen2.5-7B-Instruct-Task-12 | 1.000 | 1.000 | 0.155 | 0.987 |
| Qwen2.5-7B-Instruct-Int4 | 1.000 | 1.000 | 0.827 | 0.961 |
| Qwen2.5-7B-Instruct-Int8 | 1.000 | 1.000 | 0.914 | 0.739 |
| Qwen2.5-7B-Open-R1-Distill | 1.000 | 1.000 | 0.302 | 0.955 |
| Qwen2.5-14B | 0.324 | 0.487 | 1.000 | 1.000 |
| Qwen2.5-14B-Instruct | 0.403 | 0.702 | 1.000 | 1.000 |
| Qwen2.5-Coder-14B | 0.492 | 0.660 | 1.000 | 1.000 |
| oxy-1-small | 0.492 | 0.104 | 1.000 | 1.000 |
| Qwen2.5-14B-Gutenberg-Instruct-Slerpeno | 0.856 | 0.931 | 1.000 | 1.000 |
| Qwen-story-test-qlora | 0.112 | 0.898 | 1.000 | 1.000 |
| Qwen2.5-14B-Instruct-GPTQ-Int4 | 0.583 | 0.092 | 1.000 | 1.000 |
| DeepSeek-R1-Distill-Qwen-14B | 0.204 | 0.000 | 1.000 | 1.000 |
| Llama-3.1-8B | 0.638 | 0.714 | 0.156 | 0.000 |
| Llama-3.1-8B-Instruct | 0.643 | 0.118 | 0.128 | 0.510 |
| Llama-3.1-8B-Fireplace2 | 0.715 | 0.532 | 0.065 | 0.354 |
| Llama-3.1-8B-TLDR | 0.925 | 0.811 | 0.331 | 0.014 |
| Llama-3.1-8B-Carballo | 0.001 | 0.605 | 0.003 | 0.946 |
| Llama-3.1-8B-Instruct-Abliterated | 0.061 | 0.005 | 0.121 | 0.422 |
| Llama-3.1-8B-Instruct-HalfAbliterated-TIES | 0.356 | 0.509 | 0.796 | 0.000 |
| Llama-3.1-8B-ExtraMix | 0.640 | 0.160 | 0.721 | 0.875 |
| Llama-3.1-8B-Instruct-cv-job-description-matching | 0.684 | 0.292 | 0.004 | 0.001 |
| Llama-3.1-8B-Instruct-PsyCourse-fold7 | 0.682 | 0.057 | 0.063 | 0.191 |
| Llama-3.1-8B-Instruct-8bit | 0.798 | 0.214 | 0.154 | 0.385 |
| Llama-3.1-8B-Instruct-4bit | 0.513 | 0.023 | 0.005 | 0.933 |
| Llama-3.1-8B-Instruct-Open-R1-Distill | 0.800 | 0.101 | 0.772 | 0.024 |
| Mistral-7B-v0.3 | 0.224 | 0.352 | 0.506 | 0.044 |
| Mistral-7B-v0.3-Instruct | 0.000 | 0.000 | 0.984 | 0.104 |
| AQUA-7B | 0.014 | 0.613 | 0.703 | 0.569 |
| Mistral-7B-v0.3-Spellcheck | 0.005 | 0.481 | 0.872 | 0.105 |
| Mistral-7B-v0.3-Instruct-demi-merge | 0.904 | 0.774 | 0.058 | 0.522 |
| Mistral-7B-v0.3-Brain | 0.340 | 0.983 | 0.481 | 0.016 |
| Mistral-7B-v0.3-Instruct-GPTQ-4bit | 0.002 | 0.281 | 0.089 | 0.784 |
| Mistral-7B-distilled-from-deepseek-r1-qwen32b | 0.009 | 0.001 | 0.511 | 0.015 |
| Gemma-2-2b | 0.747 | 0.869 | 0.689 | 0.000 |
| Gemma-2-2b-it | 0.944 | 0.039 | 0.626 | 0.060 |
| Gemma-2-baku-2b | 0.616 | 0.231 | 0.176 | 0.464 |
| Gemma-2-2b-neogenesis-ita | 0.002 | 0.536 | 0.079 | 0.980 |
| Gemma-2-2b-merged | 0.951 | 0.012 | 0.651 | 0.075 |
| Gemma-2-2b-it-lora-sql | 0.236 | 0.408 | 0.000 | 0.000 |
| Gemma-2-2B-it-4Bit-GPTQ | 0.945 | 0.159 | 0.000 | 0.008 |
| Gemma-2-2b-it-distilled | 0.482 | 0.573 | 0.271 | 0.276 |
| Llama-2-7b | 0.443 | 0.470 | 0.148 | 0.276 |
| Llama-2-7b-chat | 0.528 | 0.868 | 0.424 | 0.766 |
| tulu-2-7b | 0.125 | 0.272 | 0.170 | 0.888 |
| EYE-Llama_qa | 0.657 | 0.244 | 0.674 | 0.968 |
| coma-7B-v0.1 | 0.953 | 0.004 | 0.573 | 0.731 |
| llama2-Better-Tune | 0.136 | 0.699 | 0.469 | 0.860 |
| Llama-2-7B-Chat-GPTQ | 0.594 | 0.603 | 0.582 | 0.424 |
| llama-2-7b-logit-watermark-distill-kgw-k1-gamma0.25-delta2 | 0.349 | 0.224 | 0.366 | 0.976 |

Table 15: **Sample results.** Pairwise similarity between target models (rows) and base models (columns). Cells with $1 - p \geq 0.99$ are highlighted in  green  as same-lineage pairs.

