# OpenReview forum: "SeedPrints: Fingerprints Can Even Tell Which Seed Your Large Language Model Was Trained From"
_ICLR.cc/2026/Conference — ICLR 2026 Poster_

### Official Review · Reviewer_cFVL · 2025-10-31

**Soundness:** 3
**Presentation:** 3
**Contribution:** 2
**Rating:** 4
**Confidence:** 5

**Summary:**

This paper investigates whether large language models (LLMs) inherit statistically detectable traits from their random initialization seeds, a phenomenon termed “Galtonian fingerprints.” The authors propose SeedPrints, a training-free statistical framework that traces model lineage at the seed level by measuring correlations across output distributions. Systematic experiments on 58 models spanning six major LLM families (LLaMA, Qwen, Mistral, Gemma, Phi, and OLMo) show that token preference biases established at initialization remain stable, even after diverse training and deployment interventions such as Instruct tuning, finetuning, PEFT, quantization, model merging, and distillation. These findings indicate that a model’s identity is partially determined at birth (initialization) rather than solely during training, providing a practical tool for model provenance analysis.

**Strengths:**

The paper proposes a statistical method for detecting and characterizing persistent fingerprints left by random initialization seeds in LLMs.
Specifically,  the authors introduce the novel idea that random seeds can leave stable, measurable fingerprints that persist through training. The findings reveal that seed-level fingerprints remain robust despite continued training across diverse datasets and objectives.

**Weaknesses:**

1. The paper empirically demonstrates the effect but does not fully explain why seed-level bias persists after optimization.
2. Reliance on Kendall-Tau significance may be unstable for small sample sets; sensitivity analysis (sample size vs. detection power) should be more explicit.
3. The motivation for using the Kolmogorov-Smirnov (KS) statistic is unclear.

**Questions:**

1. Could the authors provide a theoretical or intuitive explanation for why seed-induced bias persists after long training? Is it due to early optimization path dependency or weight-space geometry?
2. How many samples are required to detect a fingerprint reliably (e.g., 95% confidence)? Please quantify this relation to support practical use.
3. Please conduct experiments on Llemma-7b[c1] in Table 5, because the robustness of the proposed method against large-scale data fine-tuning needs to be verified.
4. The motivation for using the Kolmogorov-Smirnov (KS) statistic needs to be clarified. In such a setting, REEF underperforms ICS, which is contrary to common sense.



[c1] Zhangir Azerbayev, Hailey Schoelkopf, Keiran Paster, Marco Dos Santos, Stephen McAleer, Albert Q Jiang, Jia Deng, Stella Biderman, and Sean Welleck. Llemma: An open language model for mathematics. arXiv preprint arXiv:2310.10631, 2023.

---

> ### Author Response · Authors · 2025-11-24
> **(1/2) To Reviewer cFVL**
>
> We thank the reviewer for their thoughtful feedback. Below we respond to the comments in **Weaknesses (W)** and **Questions (Q)**.
>
>
> >***W1: The paper empirically demonstrates the effect but does not fully explain why seed-level bias persists after optimization.***
>
> We thank the reviewer for this insightful question! As it essentially overlaps with the first question (***“Q1: Could the authors provide a theoretical or intuitive explanation for why seed-induced bias persists after long training? Is it due to early optimization path dependency or weight-space geometry?”***), we address them together below:
>
> **1. Why such bias exists (theoretical result):** Understanding the origin of seed-level bias is essential to explaining its persistence. Our key observation is that **the bias is not random noise from initialization, but arises from a seed-dependent contraction of hidden representations**. When random inputs propagate through layers of asymmetric activations in MLP (with joint inductive bias from layer norm and self-attention module), their hidden representations contract toward a common mean direction. This direction is determined by the initialized weights, and is therefore *seed-dependent*. As a result, even before training, the output logits over random inputs already exhibit specific token-level preferences rather than a uniform distribution. Building upon observation, we actually develop a full theoretical analysis of transformers that characterizes this contraction phenomenon. The theory is **not** restricted to the fingerprinting setting, but concerns more general learning dynamics of transformers. These results are presented in a separate theoretical companion work that is currently under review; fully stating the formal theorems and proofs would require substantially more space than this conference paper allows. Here, we only summarize the high-level insights.
>
>
> **2. Why it persists:** The intuition behind the empirical persistence is two-fold. (1) Due to the high dimensionality of the parameter space, during training, not all these seed-induced directions receive optimization pressure; some (if not many) of these components are weakly correlated or nearly orthogonal to task gradients. Since SGD updates concentrate on task-relevant subspaces and modern LLMs are highly overparameterized, the optimizer can fit the data by adjusting other degrees of freedom without overwriting all seed-specific directions (recall the lottery ticket hypothesis [1]). A subset of such weakly updated components is sufficient for the bias to be observed.  (2) We deliberately use the signals from random embeddings (i.e., inputs that are likely off the training manifold) rather than natural-language tokens. This reduces interference from task-induced patterns and makes the initialization signal easier to detect statistically.
>
> ---
>
> >***W2: Reliance on Kendall-Tau significance may be unstable for small sample sets; sensitivity analysis (sample size vs. detection power) should be more explicit.***
>
> We thank the reviewer for raising this point. Because it is closely related to Q2 (***“How many samples are required to detect a fingerprint reliably?”***), we address them together here.
>
> We first clarify that the “samples” used in the Kendall’s tau test are random input sequences synthetically generated by the auditor. These inputs can be produced in arbitrarily large quantities, so extremely small sample sizes (e.g., $n \leq 30$) are not a practical constraint in our setting. Besides, since our method is formulated as a hypothesis test rather than a point estimator (e.g., estimating a single similarity score), confidence intervals are not applicable here; uncertainty is already captured through hypothesis testing by comparing two Kendall–Tau distributions. If the reviewer’s use of “confidence’’ refers instead to the significance level of the test, then by default all our experiments use $\alpha = 0.01$.
>
> **How many samples (i.e., $n$) are required**
>
> We first provide a theoretical analysis in $\\textrm{\\color{blue}Appendix C.2}$. In short, successful recovery of population-level bias requires the sample complexity of $n = \Omega \left(\log (d_{\text{out}}) / \gamma^2\right)$, where $\gamma$ is the bias margin (which increases with sequence length $l$ due to accumulation). Thus, without reaching this bound, larger $n$ always improves performance.
>
> Empirically, we use $n=2000$ (when $l=1024$) and observe stable detection performance across all tested models. We added ablation studies in $\\textrm{\\color{blue}Table 8}$. It shows that performance saturates around this setting, while smaller values (e.g., $n \le 500$) would degrade performance. That's saying, for commonly used models with $d_{\text{out}} \le 5120$, $n=2000$ appears sufficient and likely exceeds the theoretical threshold under $l=1024$.

---

> ### Author Response · Authors · 2025-11-24
> **(2/2) To Reviewer cFVL**
>
> >***W3: The motivation for using the Kolmogorov-Smirnov (KS) statistic is unclear.***
>
> This concern is closely related to Q4 (***“The motivation for using the Kolmogorov–Smirnov (KS) statistic needs to be clarified. In such a setting, REEF underperforms ICS, which is contrary to common sense.”***). We therefore address them together.
>
> **We report KS for two reasons.** First, it is the built-in metric used in LeafBench [2], which allows a fair comparison within this benchmark. Second, KS directly captures *distribution-level separability* without hinging on any particular similarity threshold. For fingerprinting, such separability is important: the goal is for lineage-consistent pairs to have score distributions that are clearly distinguishable from those of unrelated pairs.
>
>
> While separability could in principle be quantified using geometric distance measures (e.g., Mahalanobis distance) on similarity scores, SeedPrints outputs *p-values* rather than linear similarity scores. Applying geometric distances directly to p-values is ill-defined: a difference such as $p = 0.01$ vs. $0.02$ does not represent a meaningful notion of “closeness” in Euclidean or Mahalanobis sense; these values encode tail probabilities of statistical evidence, not coordinates in a metric space. What matters, instead, is how well the *distributions* of p-values for positive vs. negative pairs are separated.
>
> The KS statistic is therefore a natural choice: it measures the maximum difference between two empirical CDFs over the entire score range and thus quantifies this distributional separation. It complements AUROC, which focuses on ranking performance under threshold sweeps.
>
> **Regarding the observation that ICS outperforms REEF under KS**, we understand the intuition but do not view this as contradictory. KS and AUROC emphasize distribution margins, and ICS tends to produce more separated similarity-score distributions (Tables 6-7). By contrast, REEF performs better in Table 5, which evaluates performance based on absolute score values; its similarity scores are more concentrated. These metrics thus highlight different aspects of the score geometry rather than yielding inconsistent conclusions. Moreover, this pattern is consistent with the SoK study [2], whose comprehensive empirical results also report that REEF generally slightly underperforms ICS.
>
> ---
>
> >***Q3: Please conduct experiments on Llemma-7b in Table 5, because the robustness of the proposed method against large-scale data fine-tuning needs to be verified.***
>
> We have conducted the requested experiments on Llemma-7B and reported the results in the updated $\\textrm{\\color{blue}Table 5}$. In addition, to further verify the robustness of our method under large-scale training, we also evaluate SeedPrints on the OLMo-2-7B series, for which multiple intermediate **pretraining checkpoints (spanning trillions of tokens)** are publicly available; see $\\textrm{\\color{blue}Figure 3}$. SeedPrints remains effective in this setting as well.
>
> ---
>
> **References**
>
> *[1] Frankle, J., & Carbin, M. (2018). The lottery ticket hypothesis: Finding sparse, trainable neural networks.*
>
> *[2] Shao, Shuo, et al. "Sok: Large language model copyright auditing via fingerprinting." arXiv preprint arXiv:2508.19843 (2025).*

---

### Official Review · Reviewer_Jec7 · 2025-10-31

**Soundness:** 3
**Presentation:** 2
**Contribution:** 3
**Rating:** 6
**Confidence:** 3

**Summary:**

This paper proposes SeedPrints, a stronger and more intrinsic notion of LLM fingerprinting. The SeedPrints methodology is built on a key empirical observation: randomly initialized LLMs exhibit subtle but reproducible token selection biases that are unique to their initialization seed. These biases persist throughout training, creating a detectable signature. Experiments show that SeedPrints remains effective across all training stages and robust under domain shifts or parameter modifications.

**Strengths:**

- The idea of seed-based intrinsic fingerprinting is interesting.
- The proposed method is validated through extensive experiments.
- It demonstrates a certain degree of practical application value.

**Weaknesses:**

- Some descriptions in the paper are unclear. For example, the terms "min/max" and "top/bottom" appearing in Figure 1 are not explained.
- There is a lack of explanation as to why models with different seeds possess different "identity indices," which is somewhat counter-intuitive. Is it theoretically possible for two models, initialized with different seeds, to have statistically indistinguishable SeedPrints due to some coincidence?
- The paper does not discuss potential evasion strategies. For instance, if a malicious actor, understanding how SeedPrints work, might attempt to develop methods to "erase" the model's original fingerprint or "forge" a fingerprint to make it appear to originate from a different seed.
- Although the paper mentions its inapplicability to black-box models, it must be acknowledged that this limits its application to some extent.
- There is a typo: in line 136, "10001st" should be "1025th".

**Questions:**

- In Figure 1 (top-right subplot), why is the baseline at -0.0003?
- In lines 178-179, why is the mean output vector calculated? Will the mean output vector differ for different values of n, thereby affecting identity indices?
- In practical applications, how is the size of m determined? Do different m values affect the stability and discriminative power of the fingerprint? Should the value of m vary for different models based on their size and vocabulary?
- In Table 5, how does the t-test result change? Based on the U-test results, it seems that the correlation (if any) weakens as the number of parameters increases.
- In Algorithm 1, is it possible for no intersection of identity indices to exist?

---

> ### Author Response · Authors · 2025-11-24
> **(1/4) To Reviewer Jec7**
>
> Thank you for your supportive review and suggestions. We respond to each point below.
>
> >***W1: Some descriptions in the paper are unclear. For example, the terms "min/max" and "top/bottom" appearing in Figure 1 are not explained.***
>
> Thanks for pointing this out. We have revised the $\\textrm{\\color{blue}Figure 1}$ caption to clearly define these terms. Specifically, max/min refer to the tokens selected via argmax/argmin next-token logits, while top/bottom refer to a set of 100 tokens that most frequently appear as argmax-selected or argmin-selected tokens over random inputs, respectively.
>
> ---
>
> >***W2: There is a lack of explanation as to why models with different seeds possess different "identity indices," which is somewhat counter-intuitive. Is it theoretically possible for two models, initialized with different seeds, to have statistically indistinguishable SeedPrints due to some coincidence?***
>
> Thank you for the insightful question. While the primary focus of this paper is to empirically demonstrate this phenomenon and to leverage it for fingerprinting, we outline below several high-level intuitions explaining why seed-specific biases arise.
>
> **(1) Representation contraction induces extreme next-token preferences.**
> The extreme next-token preferences (e.g., consistently favoring specific argmax/argmin tokens) originate from a contraction of final-token representations across different random input sequences. That is, hidden representations produced from diverse random inputs collapse toward a common direction, inducing systematic bias in the output distribution.
>
> **(2) Contraction originates from asymmetric nonlinearities and normalization.**
> This inter-sequence representation contraction is primarily driven by asymmetric nonlinear activations (together with LayerNorm) in MLP blocks, which introduce seed-dependent mean shifts that propagate through the network. Such contraction is further amplified by self-attention. Different initialization seeds produce different contraction directions by influencing millions or even billions of parameters involved in forming this bias.
>
> **(3) Collisions are unlikely and empirically unobserved.**
> Since the parameter space is high dimensional, the probability that two independently initialized models exhibit statistically indistinguishable behavior is exponentially small in model dimension---especially when probed using a large number of random inputs, as in our evaluation. Empirically, across hundreds of seeds (in $\\textrm{\\color{blue}Figure 5}$ in $\\textrm{\\color{blue}Appendix B.1}$) and multiple prevailing architectures, we have not observed a single collision.
>
> All results in (1) and (2) are from a separate theoretical companion work that is currently under review; fully stating the formal theorems and proofs would require substantially more space than this conference paper allows. Therefore, we only summarize the high-level insights here.
>
> ---
>
> >***W3: The paper does not discuss potential evasion strategies. For instance, if a malicious actor, understanding how SeedPrints work, might attempt to develop methods to "erase" the model's original fingerprint or "forge" a fingerprint to make it appear to originate from a different seed.***
>
> Frankly speaking, we do not yet know what an effective evasion strategy would look like, as SeedPrints is built upon the intrinsic and systematic bias in transformer architectures (as explained in W2). We have already attempted to construct adversarial scenarios by covering four practical forms of model alteration in $\\textrm{\\color{blue}Section 5.3}$, like incorporating other models' parameters (model merging) or other models' knowledge (distillation) to confuse the detection. Our method remains effective in all these settings, while baselines such as REEF, Gradient, and ICS can be evaded.
>
> We believe that carefully-designed evasion attacks are meaningful future work, and we are happy to evaluate any potential strategies if you have specific ideas in mind.

---

> ### Author Response · Authors · 2025-11-24
> **(2/4) To Reviewer Jec7**
>
> >***W4: Although the paper mentions its inapplicability to black-box models, it must be acknowledged that this limits its application to some extent.***
>
> Thank you for the suggestion. We have now added an explicit limitations paragraph to the $\\textrm{\\color{blue}Section 6}$, and we provide a full discussion in $\\textrm{\\color{blue}Appendix A.2}$, as shown below:
>
> Although this work uncovers seed-born, training-persistent biases that can serve as fingerprint signals, our method remains within the white-box paradigm. It may not apply when only restricted API access is available; for instance, when responses are deterministic or safety-filtered. However, we emphasize that the pursuit of **true fingerprints** for LLM is still at a very early stage, even under the white-box setting. Existing fingerprinting approaches rely on training-induced signatures or alignment behaviors, and thus cannot robustly protect pretrained or early-stage models (Table 3 and Figure 3). In analogy to human fingerprints, which remain stable regardless of age or environment, we believe that a true LLM fingerprint should persist across continued training and deployment contexts rather than reflect incidental training artifacts. Achieving this--particularly under black-box access, where there remains a substantial performance gap compared to white-box methods--remains a challenging open direction and requires substantial future work. We hope this work motivates further research toward establishing principled, robust fingerprinting frameworks for large models.
>
> ---
>
> >***W5: There is a typo: in line 136, "10001st" should be "1025th".***
>
> Thanks for pointing this out. We have corrected it in the revision.
>
> ---
> >***Q1: In Figure 1 (top-right subplot), why is the baseline at -0.0003?***
>
> Thank you for noting this. The previous figure showed a **single random sample** of Kendall’s tau. We have now added the **mean ± std over 100 trials** for the random baseline. Under independence, Kendall’s tau has a theoretical standard deviation of about ($\approx 6.6\times 10^{-4}$) for our effective sample size. The original value –0.0003 lies well within this expected null range, and the correlations arising from initialization bias strictly lie above this range ([-0.0006,0.0006]).
>
> ---
>
> >***Q2: In lines 178-179, why is the mean output vector calculated? Will the mean output vector differ for different values of n, thereby affecting identity indices?***
>
> Thank you for the good question. We provide detailed theoretical explanations in $\\textrm{\\color{blue}Appendix C.1}$ and $\\textrm{\\color{blue}Appendix C.2}$. Our goal in this step is to capture the *underlying population-level bias* of the model (over the random inputs distribution). While other alternatives exist, e.g., **per-sample argmin voting** (selecting the least-value index for each input and then counting the most frequent indices), the sample mean is a more statistically stable and reliable estimator for this purpose.
>
> $\\textrm{\\color{blue}Appendix C.1}$ shows that the instability of the sample mean is **upper bounded** and **contracts as $n$ grows** due to standard concentration. Since $n$ corresponds to random inputs generated by the auditor, a large $n$ is easy to obtain, and the estimator becomes progressively more stable. $\\textrm{\\color{blue}Appendix C.2}$ further shows that once $n$ exceeds a mild lower bound (determined by the population gap and output dimension), the sample mean can **accurately recover the true underlying identity indices**.
>
> In contrast, alternatives such as **per-sample argmin voting are inherently unstable**; indeed, increasing $n$ only raises the likelihood of index flips.
>
> This is why we compute the mean output vector: it is both **stable with respect to numerical randomness** (C.1) and **provably reliable in recovering the true underlying bias** (C.2).

---

> ### Author Response · Authors · 2025-11-24
> **(3/4) To Reviewer Jec7**
>
> >***Q3: In practical applications, how is the size of m determined? Do different m values affect the stability and discriminative power of the fingerprint? Should the value of m vary for different models based on their size and vocabulary?***
>
> We thank the reviewer for raising this question.
> In practice, we set $m = 0.1 d_{\text{out}}$, which corresponds to selecting the lowest 10% mean-score coordinates. This choice works robustly across a wide range of model sizes (2B–14B) and architectures families, including Qwen, Llama, Mistral, Gemma, TinyLlama, and newly added OLMo variants ($\\textrm{\\color{blue}Section 5.2}$ and $\\textrm{\\color{blue}5.3}$). Since $d_{\text{out}}$ typically lies between 3k–5k on these foundation models, this choice is effectively equivalent to choosing $m \approx 200–500$.
>
> **Does $m$ affect performance?**
>
> Yes. We include an ablation study ($\\textrm{\\color{blue}Section 5.4}$, $\\textrm{\\color{blue}Table 8}$) and a theoretical justification ($\\textrm{\\color{blue}Appendix C.2}$). In principle, too small $m$ (e.g., $<30$) yields insufficient support to represent the distribution for hypothesis testing, thus reducing test power. Conversely, as observed, too large $m$ would introduce many weak-signal dimensions, adding noise to the coordinate set. This would shrink the bias margin ($\gamma$ in $\\textrm{\\color{blue}Corollary C.4}$), increasing the sample complexity required to capture the underlying bias.
>
> However, empirically this is not a practical concern, as a broad range of values performs well: for example, fixing $m = 200$ or $400$ (Table 8), or scaling it by $0.1 d_{\text{out}}$, all yield similarly strong results. This also answers **"Should $m$ be varied?"**: It may vary mildly with output dimension, but such variation is optional according to the empirical performance.
>
> ---
>
> >***Q4: In Table 5, how does the t-test result change? Based on the U-test results, it seems that the correlation (if any) weakens as the number of parameters increases.***
>
> We appreciate the reviewer’s question. First, we would like to clarify that in Table 5 all models have **the same parameter size (7B)**; the “# Tokens” column reports the **amount of additional training data (tokens seen)** used to obtain each derivative model, not the number of parameters.
>
> **Regarding the trends:** After examining both the U-test and t-test results, we do not observe any consistent trend indicating that the lineage signal weakens as the number of tokens increases. Importantly, the models in Table 5 differ not only in token counts but also in training domains (ranging from finance and medical to code). These heterogeneous training setups naturally introduce model-specific variability.
>
> To further validate this, we computed t-test values for all models in $\\textrm{\\color{blue}Table 5}$ and additionally included a model trained on **700B tokens** (Llemma-7B). **Across all models, the t-test p-values remain extremely small, and large-token models can even appear *more* significant (e.g., $10^{-25}$) than smaller-token ones (e.g., $10^{-15}$)**. This confirms that the SeedPrints signal remains robust across a wide range of training-token scales, with no evidence of monotonic weakening. Lastly, because p-values represent tail probabilities, comparing numbers such as $10^{-63}$ and $10^{-200}$ is not meaningful once they are already in the extreme tail region. We have included these t-test numbers in the Appendix as well.
>
>
> | Model                                   | # Tokens | t-test \(p < 0.01\)                    |
> |-----------------------------------------|----------|-----------------------------------------|
> | Llama-2-finance-7B *(Heenan, 2023)*     | 5M       | $4.13 \times 10^{-275}$ ✓             |
> | Vicuna-1.5-7B *(Chiang et al., 2023)*   | 370M     | $0.00\mathrm{E}+00$ ✓                 |
> | Wizardmath-7B *(Luo et al., 2023)*      | 1.8B     | $0.00\mathrm{E}+00$ ✓                 |
> | Meditron-7B *(Chen et al., 2023)*       | 48B      |  $2.219\times 10^{-63}$ ✓ |
> | CodeLlama-7B *(Meta AI, 2024)*          | 500B     | $2.98 \times 10^{-15}$ ✓              |
> | Llemma-7B *(Azerbayev et al., 2023)*| 700B | $8.09 \times 10^{-25}$ ✓          |

---

> > ### Author Response · Authors · 2025-11-24
> > **(4/4) To Reviewer Jec7**
> >
> > >***Q5: In Algorithm 1, is it possible for no intersection of identity indices to exist?***
> >
> > **We have not observed any case with $k = |\mathcal{M}(f)\cap \mathcal{M}(f')| = 0$ in our entire evaluation suite.** Indeed, for models from different origins we sometimes see smaller overlaps, but **an empty intersection would itself be a strong signal of non-lineage**: if two models share no identity coordinates at all, there is effectively no statistical evidence that they originate from the same initialization.
> >
> > From a probabilistic perspective, the zero-intersection scenario is extremely unlikely under the current parameterization. We set the fingerprint size $m \approx 0.1 d_{\text{out}}$ (with usually $d_{\text{out}}\in[3000,5000]$). Even if both identity sets $M(f)$ and $M(f')$ were chosen completely at random, the expected intersection size would be $\mathbb{E}[|\mathcal{M}(f)\cap \mathcal{M}(f')|] = \frac{m^2}{d_{\text{out}}}$, which is about $30$ to $50$ shared indices in our setting. The probability of an empty intersection is astronomically small; for example, with $d_{\text{out}}=4096$, $\Pr(\mathcal{M}(f)\cap \mathcal{M}(f')=\varnothing) \approx (1 - m/d_{\text{out}})^m \approx 1.8\times 10^{-19}$.
> > In practice, true descendants exhibit additional structural correlation, so their overlap $k$ is typically much larger than this random lower bound.

---

> > > ### Comment · Reviewer_Jec7 · 2025-11-27
> > >
> > > Thanks for the author's response. I will maintain my rating but increase the confidence.

---

> > > > ### Author Response · Authors · 2025-12-02
> > > >
> > > > Thank you for your follow-up and for increasing your confidence in our work. We apologize for the delayed reply — we initially planned to respond after uploading a revised version, but the system became locked shortly after. Although the discussion is now closed, we still sincerely appreciate your time and the constructive feedback that helped us strengthen the paper.

---

### Official Review · Reviewer_4uC2 · 2025-10-31

**Soundness:** 2
**Presentation:** 3
**Contribution:** 3
**Rating:** 4
**Confidence:** 4

**Summary:**

The paper identifies an interesting phenomenon in model training: randomly initialized language models exhibit some bias in their token predictions and this bias persists even after training. This signature can be leveraged to form the basis of a fingerprinting test, which can test whether a given model has statistically significant correlation with the bias detected in a given randomly initialized model. The key innovation is the introduction of _identity indices_, continuous embeddings which have more persistent signatures throughout training.

**Strengths:**

1. The paper makes use of a scientifically and technically interesting signature that is present in LLMs. To my knowledge, this is the first time this particular signature has been documented.
2. The introduction of the _identity indices_ is a clever innovation which allows the test to focus on a statistical signature which is more stable (across training).

**Weaknesses:**

1. The test itself is a bit complicated. This, in itself, is not a significant weakness, but it does makes it difficult to determine if the test is correct or not. The paper itself does not justify why the null distribution used is correct. In particular, due to the selection used to build the intersection $T$, the random variables in $\mathcal{T}$ are not independent. Thus, it seems odd to me that the null distribution treats each simulated index as being independent. I'm not saying the test is incorrect, but I do think some justification here is necessary, given the complexity of the method.

**Questions:**

1. The paper describes an identity index with a low mean $\bar{g}_j$ to be "unwilling", but in the case of the final embedding, where the dimensions are signed and there is no distinction between "true" and "non-true" classes, it's not clear why choosing with the argmin makes sense here.
2. How many parameters did the LLaMA/Qwen-style models have?
3. There is an opportunity here to show that this technique persists across scales by leveraging the checkpoints published during the training of the OLMo/OLMo 2 family of models. This is a very interesting experiment, since it shows that the signature can persists during pretraining at the scale of trillions of training tokens without having to train such models from scratch for this work.
4. In Table 2, the negative p-values are on the order of 1e-10 or larger while in Figure 2 they are on the order of 1e-50 or smaller. Why are the p-values so different in these two settings?
5. The results in Figure 1 suggest that token level biases also persist through training, although the rank correlation is quite small. Do you think this could be converted into an effective test by increasing the number of samples until the p-value becomes small enough? If it can be made to work, the token-level setting is desirable because it can be implemented without access to the continuous embeddings of the model.

---

> ### Author Response · Authors · 2025-11-24
> **(1/2) To Reviewer 4uC2**
>
> Thank you for your valuable suggestions and constructive feedback. They greatly help improve the quality of our paper. Below, we respond to each point in turn.
>
> ---
>
> >***Main weakness: Justify why the null distribution used is correct.***
>
> Thanks for the intriguing question! We agree that the construction of the null deserves a more explicit explanation. In the revision, we add a step-by-step justification in $\\textrm{\\color{blue}Appendix B}$. We summarize the key points below.
>
> **(1) The empirical null for independent models is Gaussian.**
> Our null aims to model the behavior of two independent models evaluated on random inputs. The strictest instantiation is two randomly initialized models. In $\\textrm{\\color{blue}Figure 3}$ (Appendix B), we compute correlations over about 2500 such pairs and show that their empirical distribution closely follows a Gaussian.
>
> **(2) Why use independent Gaussians as a proxy of model outputs.**
> Model outputs (e.g., logits or hidden activations) are commonly modeled as Gaussian in auditing and theoretical analyses [1,2,3]. Under random initialization, two models have independent parameter matrices, so their outputs are (ideally) independent given inputs. Therefore, sampling two independent Gaussian matrices is a natural proxy for random-initialized models’ outputs. While selecting $\mathcal{T}$ could induce certain level of dependence, applying the same selection procedure to Gaussian samples preserves this feature, so it does not affect the validity of the null.
>
> **(3) Why we omit $\mathcal{T}$ selection in Algorithm 1**
> In Algorithm 1, we do not re-apply $\mathcal{T}$ selection in constructing the null. This is because, empirically, we observe that the raw Gaussian null (with selection) and the simplified version (without selection) yield indistinguishable distributions ($\\textrm{\\color{blue}Figure 4}$ in Appendix B). Thus, we adopt the simplified null for efficiency without affecting results, and the comprehensive experiments in $\\textrm{\\color{blue}Section 5}$ further justify this simplification.
>
> ---
>
> >***Q1: The paper describes an identity index with a low mean g to be "unwilling", but in the case of the final embedding, where the dimensions are signed and there is no distinction between "true" and "non-true" classes, it's not clear why choosing with the argmin makes sense here.***
>
> Thanks for the question. Our motivation experiments initially use logits because they provide a more intuitive visualization. The “willing/unwilling” phrasing was meant only as an informal description of tokens that consistently receive higher or lower logit values, not as a semantic interpretation of embedding coordinates. The fingerprint signal does not rely on such semantics; rather, it reflects systematic outlier dimensions in the output distribution. **The same pattern appears in the final embedding, since logits are obtained via a linear projection from that space.**
>
> **Using embeddings is more efficient and robust in high-vocabulary settings.** We provide a formal theoretical analysis in $\\textrm{\\color{blue}Appendix C 2}$. In short, the number of random inputs $n$ required to reliably capture the underlying bias scales with the output dimensionality $d_{\text{out}}$. Since hidden dimensions are relatively small and stable across model scales (e.g., $\approx$ 7K even in the 671B DeepSeek-V3), whereas vocabulary sizes can exceed 100K, operating directly on logits is substantially more expensive under the same bias signal. We have clarified this distinction in the revision.
>
> ---
>
> >***Q2: How many parameters did the LLaMA/Qwen-style models have?***
>
> The LLaMA/Qwen-style models that we pretrain ourselves are small proof-of-concept models with roughly **160M parameters**. Their sole purpose is to demonstrate seed-level distinguishability when we do not have access to the exact initialization configuration of large open-source models. All main results on realistic large models use standard sizes: Figure 3 and Table 5 reports experiments on **7B-parameter** models, and Tables 6-8 further includes models ranging from **2B to 14B** parameters.
>
> ---
>
> **References**
>
> *[1] Carlini, N., Chien, S., Nasr, M., Song, S., Terzis, A., & Tramer, F. (2022, May). Membership inference attacks from first principles.*
>
> *[2] Tang, C., Liu, J., Xu, H., & Huang, L. (2024). Top-$ n\sigma $: Not All Logits Are You Need.*
>
> *[3] Lee, J., Xiao, L., Schoenholz, S., Bahri, Y., Novak, R., Sohl-Dickstein, J., & Pennington, J. (2019). Wide neural networks of any depth evolve as linear models under gradient descent.*

---

> ### Author Response · Authors · 2025-11-24
> **(2/2) To Reviewer 4uC2**
>
> >***Q3: There is an opportunity here to show that this technique persists across scales by leveraging the checkpoints published during the training of the OLMo/OLMo 2 family of models. This is a very interesting experiment, since it shows that the signature can persists during pretraining at the scale of trillions of training tokens without having to train such models from scratch for this work.***
>
> Thanks for the suggestion! This is indeed a strong complementary experiment to our Figure 2 results. We have added it as Figure 3 in the revision and included comparisons with all baselines. The results show that our method provides reliable verification across the full pretraining trajectory of OLMo 2 7B model (from 5B to 3.9T tokens), whereas existing baselines fail in the early stages (e.g., ICS$\approx 0$, PCS$< 0.3$ and REEF$<0.6$ within the first **trillion** tokens).
>
> ---
>
> >***Q4: In Table 2, the negative p-values are on the order of 1e-10 or larger while in Figure 2 they are on the order of 1e-50 or smaller. Why are the p-values so different in these two settings?***
>
> Good catch! The difference arises because Figure 2 starts from the 4K-step checkpoint, where even minimal data exposure already strengthens lineage signals, yielding smaller p-values. In contrast, Table 2 evaluates fully initialized models with no data knowledge at all, which is the most challenging setting; signals are naturally much weaker.
>
> This difference reinforces our key point: **prior work evaluates almost exclusively on finetuning-stage models (e.g., LLaMA2-7B families or Qwen families), where detection is comparatively easy.** In contrast, verifying lineage in the pretraining stage—especially very early in training or at initialization (“at birth”)—is substantially harder and largely underexplored. Our results show that methods that appear effective in these easier fine-tuning regimes fail under early-stage distribution shifts (e.g., $\\textrm{\\color{blue}Table 3}$ and $\\textrm{\\color{blue}Figure 3}$). Therefore, evaluation should pay more attention to the pretraining stage, where the task is more challenging and meaningful.
>
>
> ---
>
> >***Q5: The results in Figure 1 suggest that token level biases also persist through training, although the rank correlation is quite small. Do you think this could be converted into an effective test by increasing the number of samples until the p-value becomes small enough? If it can be made to work, the token-level setting is desirable because it can be implemented without access to the continuous embeddings of the model.***
>
> Thank you for the suggestion. The token-level setting is indeed desirable and your intuition is largely correct. However, the short answer is *“yes in principle, but difficult to scale”*:
>
> **It can work.** Token-level signals are shown to be effective in small-vocabulary settings. In $\\textrm{\\color{blue}Tables 1-4}$ on 160M models (vocabulary $\approx$ 32k), we successfully perform token-level tests (using logits as outputs) with $n = 10{,}000$ samples across all experiments.
>
> **Scaling is difficult.** The number of samples required to accurately recover the underlying bias scales with the output dimensionality (as previously noted in Q1; cf. $\\textrm{\\color{blue}Appendix C.2}$). For modern foundation models, the vocabulary size commonly exceeds 100k, meaning the required sample size will grow significantly (we have not yet identified the exact threshold). In contrast, the hidden size is much smaller and remains stable across scales (e.g.,  $\approx$ 7K even in the 671B DeepSeek-V3), making the hidden-state setting far more practical.
>
> However, we still view token-level detection as a meaningful direction for future work. Our current method relies on rank-based correlation, which is not optimal. Improved estimators that increase the bias margin $\gamma$ could quadratically reduce the required sample size (see $\\textrm{\\color{blue}Corollary C.4}$ in Appendix).

---

> > ### Author Response · Authors · 2025-12-02
> >
> > Thank you for having raised your score and for your support immediately after our rebuttal was posted. Because score updates on OpenReview do not trigger notifications, we only noticed it just now while reviewing the archived log in preparation for our final summary, and we apologize for the delayed acknowledgement. It is unfortunate that, due to the special circumstances this year, we cannot confirm whether all concerns were fully resolved, but we hope our revisions addressed the key points. We sincerely appreciate your contribution to improving the paper — especially the suggestion to evaluate on the OLMo family, which provided a valuable perspective and significantly strengthened our motivation for highlighting the importance of the pre-training stage.

---

### Official Review · Reviewer_2xKH · 2025-11-01

**Soundness:** 3
**Presentation:** 3
**Contribution:** 3
**Rating:** 4
**Confidence:** 4

**Summary:**

SeedPrints fingerprints LLMs using their random initialization rather than training artifacts. Untrained networks already show reproducible token selection biases set by the seed, and these signatures persist through training. A simple statistical test recovers model lineage with high confidence and stays robust across training stages, domain shifts, and moderate parameter edits, with strong results on LLaMA and Qwen families and public benchmarks.

**Strengths:**

1. Finding of the biometric fingerprint is interesting, where initialization creates stable token selection biases that are seed-specific and persist throughout training, giving each model a seed-level identity.
2. The authors find an effective algorithm by constructing identity indices from the most down-weighted output dimensions on random inputs, then run a rank-based correlation test with calibrated p-values to decide shared lineage.
3. The experiment setting is comprehensive, which verifies lineage at every checkpoint and stays accurate under domain shift, finetuning, PEFT, quantization, merging, and distillation, outperforming similarity-based baselines.

**Weaknesses:**

1. Within the same model family, how similar are fingerprints across sizes. For example, what is the similarity between Qwen2.5 7B and 14B, and does similarity grow with parameter sharing or architectural reuse, and will they be distinguishable?

2. For models like deepseek distill llama or deepseek distill qwen, which fingerprint do they most resemble? The teacher, the student base, or a new hybrid, and does SeedPrints remain reliable in these distillation settings

3. The SeedPrints relies on logits or hidden states, which many closed-source APIs do not expose. Are there practical black box surrogates to test for seed-level fingerprints, for example, using only top-k outputs or sampled continuations with controlled prompts

4. Rank-based tests may be sensitive to large vocabularies and temperature scaling. Could authors add more analysis of sensitivity to temperature, alternative normalizations before ranking, and report how these choices affect power and false positive rates

5. In the experiment, the comparisons emphasize p-values. Maybe add more effect sizes, confidence intervals for Kendall tau, could improve the interpretability and calibration

**Questions:**

Please see the weakness part. I will consider raising the score if the authors address the weaknesses/problems clearly

---

> ### Author Response · Authors · 2025-11-24
> **(1/3) To Reviewer 2xKH**
>
> Thank you for your constructive review and suggestions. Below we respond to the comments in **Weaknesses (W)**.
>
> ---
>
> >***W1: Within the same model family, how similar are fingerprints across sizes. For example, what is the similarity between Qwen2.5 7B and 14B, and does similarity grow with parameter sharing or architectural reuse, and will they be distinguishable?***
>
> We thank the reviewer for raising this question about cross-size behavior within the same model family.
>
> **(1) Are models within the same family distinguishable? New experiment on the Qwen-2.5 family**
>
> As suggested, we evaluate Qwen-2.5-14B as the target and treat Qwen-2.5-{0.5B, 1.5B, 3B, 7B} as suspect base models. The resulting scores are:
>
> | Base Model (Target: Qwen-2.5-14B) | SeedPrint $\uparrow$     | REEF $\downarrow$     | PCS $\downarrow$       |
> |-----------------------------------|---------------|---------:|-------------:|
> | Qwen-2.5-0.5B                     | 0.7184        | 0.8250   | -6.366e-05   |
> | Qwen-2.5-1.5B                     | 0.9707        | 0.8399   | -2.837e-05   |
> | Qwen-2.5-3B                       | 0.3898        | 0.8833   | -8.702e-06   |
> | Qwen-2.5-7B                       | 0.5429        | 0.8461   | -6.080e-06   |
>
> For clarity, we briefly recap how the similarity-based baselines and our method should be interpreted:
>
> 1. REEF and PCS (similarity-based baselines). They output a continuous similarity score, and the user must manually choose a threshold to decide when "similar" is "similar enough" to treat two models sharing lineage. (Different deployments may choose different thresholds, so the method itself does not provide a built-in decision rule.)
> 4. SeedPrints (hypothesis-testing view). SeedPrints instead performs a statistical hypothesis test and reports a p-value. At a **significance level** of, e.g., 0.01, we declare two models to share lineage (the target model is trained from the base model). Importantly, the absolute value of the p-value beyond this threshold does not provide a simple “more similar / less similar” interpretation (it is not a cosine similarity).
>
> **Interpreting the results:** All SeedPrints scores for Qwen-2.5-{0.5B, 1.5B, 3B, 7B} vs Qwen-2.5-14B are larger than 0.01, meaning that SeedPrints **does not assert that Qwen-2.5-14B shares the same lineage as any of these smaller models**. Therefore, the answer is: **Yes. SeedPrints can distinguish models within the same family.**
>
> **(2) Regarding the “how similar” question: SeedPrints does not measure similarity**
>
> SeedPrints addresses the binary detection problem (“same lineage vs. not”) rather than a continuous similarity–scoring problem. This actually is a key advantage of SeedPrints:
>
> * Much of the LLM fingerprinting literature is motivated by detecting model theft, unauthorized reuse, or copyright infringement of expensive proprietary models. In such high-stakes scenarios, model owners typically need a clear yes/no verdict about whether a suspect model is plausibly derived from a protected base model, not just a relative similarity score.
> * Similarity-based methods such as REEF and HuRef only produce relative scores and require users to choose heuristic thresholds or calibrate against a set of proxy models drawn from a similar model distribution. This makes intermediate scores difficult to interpret and often brittle across different operating regimes.
> * In contrast, SeedPrints provides a principled statistical decision at a fixed significance level, yielding a clear yes/no outcome on whether a candidate exhibits initialization-born fingerprints consistent with the target model.
>
> Overall, we found the reviewer’s question very constructive and helpful in sharpening our discussion. We will include this Qwen-2.5 cross-size experiment and its interpretation in the Appendix.

---

> ### Author Response · Authors · 2025-11-24
> **(2/3) To Reviewer 2xKH**
>
> >***W2: For models like deepseek distill llama or deepseek distill qwen, which fingerprint do they most resemble? The teacher, the student base, or a new hybrid, and does SeedPrints remain reliable in these distillation settings***
>
> We thank the reviewer for this question about distillation. In the Distillation column of $\\textrm{\\color{blue}Table 7}$, we already evaluate several realistic distillation cases, including DeepSeek-R1-Distill-Qwen-14B (Please check the $\\textrm{\\color{blue}Appendix C 1.2}$ for model usage). The distilled model is compared against a pool of candidate bases, SeedPrints attributes it to the correct student base.
>
> To directly address the “teacher vs. student” aspect on a pair where both sides are accessible, we further ran a controlled experiment with (Note because the teacher - full deepseek 671B model used for distilling is too large for us to host on our school-level server, we use the following setting):
>
> - Teacher: Gemma-2-9B-it
> - Student base: Gemma-2-2B-it
> - Distilled student: Gemma-2-2B-it-distilled
>
> Treating the distilled model as the suspect, SeedPrints yields the following p-values ($\alpha$ = 0.01):
>
> | Base model (suspect: Gemma-2-2B-it-distilled) | t-test p-value | u-test p-value |
> | --------------------------------------------- | -------------- | -------------- |
> | Gemma-2-9B-it (teacher)                       | 0.1893         | 0.3551         |
> | Gemma-2-2B-it (student base)                  | 3.398e-168     | 6.756e-55      |
>
> Under our decision rule (p < 0.01 => same lineage), SeedPrints clearly **matches the distilled model with the student base, not the teacher**. This is correct. Conceptually, in the setting, the teacher just offline generate more data for training, while the student’s parameters are initialized and updated within the student architecture.
>
> ---
>
> >***W3: The SeedPrints relies on logits or hidden states, which many closed-source APIs do not expose. Are there practical black box surrogates to test for seed-level fingerprints, for example, using only top-k outputs or sampled continuations with controlled prompts***
>
> We appreciate the reviewer for raising this point. Our current method indeed assumes a white-box setting with access to logits or hidden states, and we do not claim that SeedPrints in its present form is directly applicable to typical API-only closed-source models
>
> **Actually, even in the pure white-box setting, how to perform fingerprinting is still an open problem.**
>
> To anwer this question properly, it is helpful to step back and reconsider what “LLM fingerprinting” means. The term fingerprint in this line of work was originally borrowed from biological fingerprints, as formalized by Francis Galton’s Finger Prints (1892): **a fingerprint is a unique and stable identifier established at birth and preserved throughout a person’s life**. By analogy, for LLMs the natural notion of “birth” is the random initialization. *A genuine LLM fingerprint, in this Galtonian sense, should allow us—given a trained model and a candidate set of randomly initialized models with different seeds—to recover which initialization seed it originated from, even after substantial training.*
>
> Most prior methods focus on training- or alignment-stage artifacts (e.g., particular data, RLHF quirks, self-reported model identity), which are often fragile under long continued training ($\\textrm{\\color{blue}Table 3}$) or heavy fine-tuning ($\\textrm{\\color{blue}Table 5}$). To further illustrate this, we additionally include experiments on the OLMo-2-7B series in $\\textrm{\\color{blue}Figure 3}$, for which multiple intermediate pretraining checkpoints (spanning trillions of tokens) are publicly available. We find that **all existing baselines fail at the first trillion-token stage**. From this perspective, even in the pure white-box regime, the core problem “can we recover a model’s initialization identity after training?” remains largely unsolved.
>
> SeedPrints is **intended as a first step toward restoring this stricter notion of fingerprint**. We show that there exist initialization-induced biases that (i) are present already at random init, (ii) persist through training, and (iii) can be used to reliably match a trained model back to its seed among multiple candidates.
>
> We fully agree that building practical black-box surrogates for such a fingerprints—e.g., based on top-$k$ distributions or sampled continuations under carefully controlled prompts—is an important and strictly harder follow-up problem. We think this is a very promising research direction (and sampled continuations in particular are a natural candidate), but it is beyond the scope of the current work.

---

> ### Author Response · Authors · 2025-11-24
> **(3/3) To Reviewer 2xKH**
>
> > ***W4: Rank-based tests may be sensitive to large vocabularies and temperature scaling. Could authors add more analysis of sensitivity to temperature, alternative normalizations before ranking, and report how these choices affect power and false positive rates***
>
> We thank the reviewer for raising this concern. We believe part of the confusion stems from a misunderstanding of the role of logits in our method. SeedPrints operates directly on raw logits (pre-softmax values) or final-layer hidden states. These are standard signal sources in model auditing and do not depend on the decoding temperature used at inference time. In our pipeline, we perform a single forward pass and read out internal activations; no temperature scaling is applied when computing our statistic.
>
> To address the concern about vocabulary size, we additionally perform ablations on different set of open-source models (used in Section 5.3) grouped by their vocabulary sizes. Across commonly used vocabulary scales, detection accuracy and false positive rates remain essentially unchanged, suggesting that the signal does not rely on a particular tokenization granularity:
>
> | Vocab Size |   ACC   |  FPR  |
> |-----------:|:-------:|:-----:|
> |     32000  | 1.0000  | 0.000 |
> |     32768  | 1.0000  | 0.000 |
> |    128256  | 0.9936  | 0.000 |
> |    151665  | 1.0000  | 0.000 |
> |    256000  | 1.0000  | 0.000 |
>
> We will include this analysis and table in the appendix to make the robustness to vocabulary size explicit.
>
> ---
>
> > ***W5: In the experiment, the comparisons emphasize p-values. Maybe add more effect sizes, confidence intervals for Kendall tau, could improve the interpretability and calibration***
>
> Thank you for raising this important point. Effect sizes and confidence intervals are not applied here because SeedPrints is not a similarity-based method performing point estimation. Unlike ALL prior approaches that estimate a similarity score and require calibrated thresholds, SeedPrints performs a **hypothesis test over empirical distributions** of Kendall’s tau obtained from random inputs. The p-value is therefore not a similarity score, but statistical evidence for rejecting or failing to reject shared lineage. **Uncertainty is already quantified within this hypothesis-testing framework, yielding a principled binary decision without threshold calibration.**
>
> In other words, in this formulation, tau is not simplified into a single scalar parameter (e.g., a “true” tau) to be estimated; it is a random variable induced by auditor-controlled sampling, and its empirical distribution is the object of inference. Confidence intervals are meaningful only when tau is treated as a point estimate, which is not the case here. *In fact, this distinction is one of the key contributions of our work; please refer to our previous reply to W1.*
>
> ---
>
> Finally, while some of the sensitivity questions do not directly apply to SeedPrints, we appreciate your broader interest in robustness. To facilitate deeper understanding of our method, we include both empirical and theoretical sensitivity analyses covering three key hyperparameters (i.e., number of random inputs $n$, sequence length $l$, and fingerprint set size $m$) in $\\textrm{\\color{blue}Section 5.4}$ and $\\textrm{\\color{blue}Appendix C.2}$ in the revision as a supplement.

---

### Author Response · Authors · 2025-12-02
**Final Author Summary**

We understand that this year’s ICLR process has been unusually challenging, and we appreciate the time you spent reviewing our rebuttal and additional experiments. We also thank all reviewers for their careful and constructive feedback.

We would like to note that, before the unexpected closure of the reviewer–author discussion, Reviewer 4uC2 raised the overall score to 6 on 25 Nov 2025, 01:59 UTC, and Reviewer Jec7 acknowledged our responses and increased the confidence on 27 Nov 2025, 12:20 UTC, both reflecting a positive stance toward our revisions.

Because reviewers' comments touch on different aspects of our work, below we summarize the paper’s main flow and the key revisions that strengthen it; detailed point-by-point responses appear in the individual replies.

---

> ## **Summary of SeedPrints**

***1. Existing methods fail on pretraining-stage lineage.***
Prior fingerprinting approaches (REEF, ICS, HuRef, etc.) are almost exclusively evaluated on finetuned models; we show that they misclassify lineage under early pretraining distribution shifts (Table 3) and in the newly added OLMo-2 7B long-trajectory experiments (Figure 3, suggested by Reviewer 4uC2).

***2. We propose a stricter, “Galtonian” definition of LLM fingerprints***
We explicitly argue that an LLM fingerprint should behave like a biological fingerprint: it should be **determined at birth** and **persist throughout all training and deployment**, rather than only reflecting finetuning artifacts.

***3. We identify a concrete initialization-induced contraction bias as an intrinsic fingerprint.***
Each seed produces a distinct contraction pattern in the final hidden states/logits under random inputs. The resulting seed-level fingerprint (i) separates seeds, (ii) remains stable throughout pretraining (where baselines fail), and (iii) holds under practical deployments such as finetuning, PEFT, quantization, merging, and distillation. Our experiments **span from 160M toy models to 2B–14B foundation models across all major families**, including **Qwen, Llama, Mistral, Gemma, TinyLlama**, and newly added **OLMo** variants.

***4. Unlike relative similarity scores, SeedPrints performs a hypothesis test that outputs a p-value and a deterministic decision.***
Instead of estimating a similarity score that requires post-hoc thresholding or calibration on proxy models, SeedPrints formulates lineage detection as a statistical hypothesis test over the distributions of Kendall’s τ. This yields a calibrated p-value and a yes/no decision at a fixed significance level, making uncertainty explicit and removing the need for heuristic threshold tuning.

---

> ## **Key revisions/responses**

***1. Explain the origin and persistence of the initialization-induced bias (Jec7, cFVL).***
In our responses to [Jec7](https://openreview.net/forum?id=Kan6Z0zzZi&noteId=szZDAYnLzV) and [cFVL](https://openreview.net/forum?id=Kan6Z0zzZi&noteId=4oX2UnT8I0), we explicitly explain why contraction occurs, why its direction is seed-dependent, and why these biases persist through training rather than being overwritten.

***2. Provided theoretical justification for the identity-index selection method (Jec7).***
We added a theoretical explanation showing that taking the mean output over random inputs provides a stable estimator of the underlying bias as the number of inputs increases, whereas alternative approaches such as per-sample argmin voting are unstable (Appendix C.1 and C.2). See the [response to Reviewer Jec7](https://openreview.net/forum?id=Kan6Z0zzZi&noteId=qmBNcVWHNr) for a concise summary.

***3. Added a step-by-step justification of the null distribution design ([4uC2](https://openreview.net/forum?id=Kan6Z0zzZi&noteId=IeZvUtRQKE)).***
We added Appendix B to explain:
(1) the empirical null distribution for independent models is Gaussian;
(2) independent Gaussians are an appropriate proxy for independent model outputs; and
(3) omitting the $\mathcal{T}$ step in Algorithm 1 does not affect test validity.

***4. Added theoretical and empirical analysis of hyperparameter influence (2xKH, Jec7, cFVL).***
We added theoretical analysis (Appendix C.2) and empirical ablations (Section 5.4) on the hyperparameters in SeedPrints (i.e., random input count, sequence length, vocabulary size, and fingerprint set size).

***5. Added long-training experiments in both pretraining and finetuning regimes, as well as additional practical interference scenarios.***
Following reviewer suggestions, we incorporated:
- Lineage detection over OLMo-2-7B pretraining checkpoints (5B–3.9T tokens) in Figure 3, (4uC2)
- Llemma-7B (700B-token training) as a large-scale finetuning test in Table 5, (cFVL)
- Cross-size evaluations within the Qwen-2.5 family (0.5B, 1.5B, 3B, 7B vs. 14B), (2xKH)
- Gemma-2 teacher–student distillation experiments. (2xKH)

We hope this summary helps clarify the contributions and revisions, and we appreciate your careful assessment of our work.

---

### Meta-Review · Area_Chair_1Wgd · 2026-01-07

**Summary:**

This paper introduces SeedPrints, a fingerprinting method that traces LLM lineage using seed-dependent biases originating from random initialization and persisting through training. The method uses a statistical hypothesis test over correlations on selected identity indices and is evaluated across various scenarios.

Overall, reviewers agreed that the paper is interesting and the core idea is novel. The authors also made a fair effort to address most reviewer's concerns. Still, some issues remain only partially addressed, including a deeper explanation of why the bias persists, discussion of evasion or forgery strategies, and applicability in black-box settings.

Therefore, AC is recommend acceptance.

**Reviewer Concerns:**

Concerns addressed in the rebuttal
- Justification about null distribution (4uC2): Addressed by adding an justification of the Gaussian null, empirical validation with independently initialized models, and an explanation of why the simplified null closely matches the full procedure.
- Lineage verification across the full pretraining trajectory (4uC2): Addressed by adding OLMo-2-7B checkpoint experiments.
- Question about using mean output vector (Jec7): Addressed by adding a theoretical justification.
- Hyperparameter sensitivity / sample complexity (2xKH, cFVL, Jec7): Addressed by adding ablation studies and a theoretical scaling discussion.
- Large-scale finetuning robustness (cFVL): Addressed by adding experiments on Llemma-7B.
- Distinguishability under within-family models / distillation-models (2xKH): Addressed by adding cross-size experiments within the Qwen-2.5 family models; adding controlled teacher–student distillation experiments on Gemma-2-9B-it.

Concerns still outstanding or only partially addressed
- Deeper explanation for why the bias persists: The rebuttal gives plausible intuition, but does not include a concrete theoretical analysis or direct empirical evidence.
- Discussion about evasion strategies (Jec7): The authors acknowledge the concern but do not provide concrete discussion about evasion strategies.
- Black-box applicability (2xKH, Jec7): The authors acknowledge the concern.

**Reviewer Scores:**

According to the author's summary, Reviewer 4uC2 raised the score to 6 (claimed by the authors). Reviewer Jec7 explicitly stated "will maintain my rating but increase the confidence" (claimed by the authors). For Reviewer 2xKH and Reviewer cFVL, I think that either maintaining or slightly increasing the score would be reasonable given the rebuttal and added clarifications.

---

### Decision · Program_Chairs · 2026-01-26

Accept (Poster)